# LRET-derived HADDOCK structural models describe the conformational heterogeneity required for DNA cleavage by the Mre11-Rad50 DNA damage repair complex

**Marella D Canny, Michael P Latham\***

Department of Chemistry and Biochemistry, Texas Tech University, Lubbock, United States

**Abstract** The Mre11-Rad50-Nbs1 protein complex is one of the first responders to DNA double-strand breaks. Studies have shown that the catalytic activities of the evolutionarily conserved Mre11-Rad50 (MR) core complex depend on an ATP-dependent global conformational change that takes the macromolecule from an open, extended structure in the absence of ATP to a closed, globular structure when ATP is bound. We have previously identified an additional 'partially open' conformation using luminescence resonance energy transfer (LRET) experiments. Here, a combination of LRET and the molecular docking program HADDOCK was used to further investigate this partially open state and identify three conformations of MR in solution: closed, partially open, and open, which are in addition to the extended, apo conformation. Mutants disrupting specific Mre11-Rad50 interactions within each conformation were used in nuclease activity assays on a variety of DNA substrates to help put the three states into a functional perspective. LRET data collected on MR bound to DNA demonstrate that the three conformations also exist when nuclease substrates are bound. These models were further supported with small-angle X-ray scattering data, which corroborate the presence of multiple states in solution. Together, the data suggest a mechanism for the nuclease activity of the MR complex along the DNA.

**\*For correspondence:**
michael.latham@ttu.edu

**Competing interest:** The authors declare that no competing interests exist.

## Editor's evaluation

This study on the Mre11 and Rad50 proteins is of interest to biologists studying DNA repair. Advances in the understanding of how structural states of Mre11-Rad50 complex are linked to DNA end detection and DNA processing, as addressed in this study, are of central importance to research on genome stability and DNA repair, with implications in human disease such as cancer and immune disorders. Enzymatically, RAD50 is an ATPase and MRE11 is a nuclease with both exo- and endonuclease activities. How all these functions are catalyzed by the complex remains unresolved. Through the combination of biophysical analyses and biochemical activity assays, this study identifies three conformations of ATP-bound *P. furiosus* Mre11-Rad50 complex – open, partially open, and closed – and links these activities to the Mre11-Rad50 function.

## Introduction

Mre11-Rad50-Nbs1 (MRN) is an essential protein complex required for the repair of DNA double-strand breaks (DSBs) (*Paull, 2018*; *Syed and Tainer, 2018*). This complex recognizes the broken

DNA and begins processing the break via Mre11 exo- and endonuclease activities and Rad50 ATP binding and hydrolysis (*Paull, 2018*). Nbs1, found only in eukaryotes, further modulates MR activity and signals downstream repair effectors to the site of the break (*Deshpande et al., 2016*; *Oh et al., 2016*). If DNA DSBs are not repaired, the cell may undergo cell death via apoptosis, or, if the break is not repaired correctly, a loss of genetic information or gross chromosomal rearrangements can occur, potentially resulting in immunodeficiencies and cancer (*Ciccia and Elledge, 2010*; *Oh and Symington, 2018*). Many mechanistic and structural studies have been performed on the evolutionarily conserved $Mre11_2$-$Rad50_2$ (MR) core complex from bacteria, archaea, and eukaryotes and have shown that the complex undergoes a dramatic ATP-induced global conformational change that is required for its various functions. This two-state model, which originated from X-ray crystallographic studies (*Lafrance-Vanasse et al., 2015*; *Lammens et al., 2011*; *Lim et al., 2011*; *Möckel et al., 2012*), has MR transforming from an extended arms-open-wide conformation, where the two Rad50 nucleotide-binding domains (NBDs) are far apart in space, to a 'closed' conformation that sandwiches two ATPs between the NBDs in a more compact, globular structure (*Figure 1A*). In the closed conformation, the Mre11 nuclease active sites are occluded and Rad50 can bind DNA (*Liu et al., 2016*; *Möckel et al., 2012*; *Rojowska et al., 2014*; *Schiller et al., 2012*; *Seifert et al., 2016*). Once Rad50 hydrolyzes the bound ATPs, the complex returns to the extended conformation where DNA substrates can again access the Mre11 active sites. Critically, ATP binding and hydrolysis, and therefore the cycling between states, appear to be required for DNA unwinding (*Cannon et al., 2013*), processive Mre11 nuclease activity (*Herdendorf et al., 2011*; *Paull and Gellert, 1998*), and downstream signaling events through ATM kinase (*Cassani et al., 2019*; *Lee et al., 2013*).

Because the Mre11 active site is occluded in the closed conformation, it was hypothesized that MR nuclease activity originates from either the extended or an otherwise unknown intermediate structure (*Deshpande et al., 2014*; *Lafrance-Vanasse et al., 2015*; *Lammens et al., 2011*; *Möckel et al., 2012*). A recent cryo-EM structure of *Escherichia coli* MR (called SbcCD) bound to a double-stranded DNA (dsDNA) substrate revealed a structure where the ADP-bound Rad50s are associated, and the Mre11 dimer has moved to one side to interact asymmetrically with the two Rad50s and the dsDNA substrate (*Käshammer et al., 2019*). In addition, we have previously used luminescence resonance energy transfer (LRET) experiments to illuminate the presence of a 'partially open' conformation in a truncated construct of hyperthermophilic *Pyrococcus furiosus* MR (Pf MR^NBD) in both ATP and ATP-free conditions (*Boswell et al., 2020*). Thus, a variety of functionally relevant structures of the MR complex may exist in solution. As LRET has been successfully used to characterize the interactions of NBDs in several ABC ATPase membrane proteins (*Cooper and Altenberg, 2013*; *Zoghbi et al., 2017*; *Zoghbi et al., 2012*; *Zoghbi and Altenberg, 2018*), we significantly extended our initial LRET studies on the Pf MR^NBD complex to further characterize this partially open conformation. Multiple LRET probes were introduced throughout the Rad50 NBD to determine a network of distances between residues across the Rad50-Rad50 interface (*Figure 1B*). These LRET-determined distances were then used as unambiguous distance restraints in the molecular docking program HADDOCK (*Dominguez et al., 2003*; *van Zundert et al., 2016*) to obtain models of the ATP-bound MR^NBD complex. Here, we present structural models of three distinct conformations of the ATP-bound Pf MR^NBD complex: closed, partially open, and open. We also demonstrate that these three states exist regardless of nucleotide state and in the presence of model DNA substrates. LRET experiments on full-length Pf MR, where Rad50 contains the coiled-coil domains and an apical zinc hook dimerization motif, confirmed that these conformations are also observed for the complete MR complex. Site-directed mutagenesis was used to disrupt specific conformations, and the effects on Mre11 and Rad50 activities validated our models and put them into a functional context. Finally, small-angle X-ray scattering (SAXS) was also employed to confirm the presence of multiple conformations of MR^NBD in solution and to assign approximate populations to each. In conclusion, we have combined orthogonal biophysical and computational methods to describe three distinct global conformations of Pf MR in solution and demonstrate that they offer valuable insight into how MR functions along the DNA.

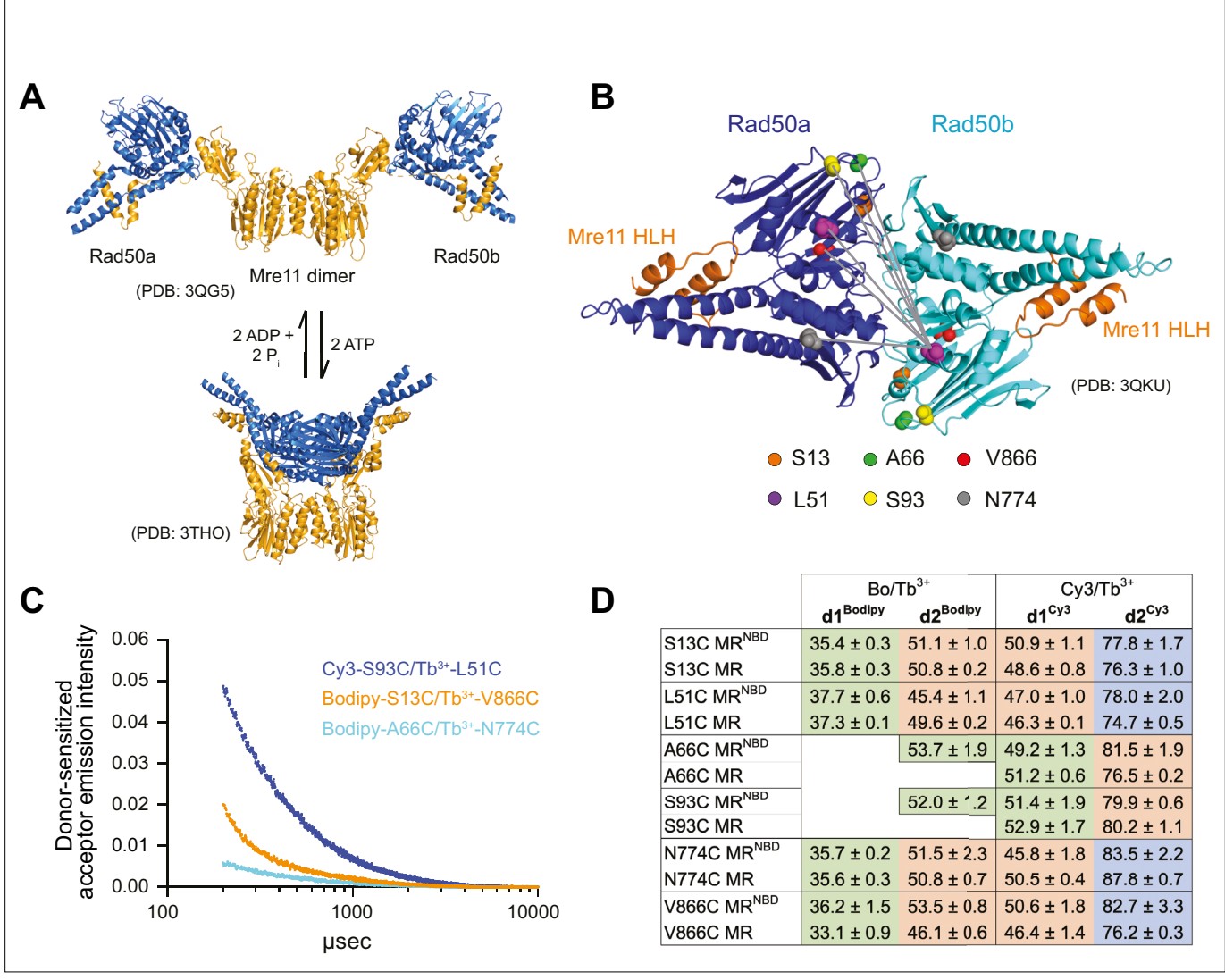

**Figure 1.** Luminescence resonance energy transfer (LRET) measures distances between Rad50 residues in Pf MR[NBD]. (**A**) X-ray crystal structures of *T. maritima* MR[NBD] showing the ATP-dependent transition between extended and closed conformations (*Lammens et al., 2011*; *Möckel et al., 2012*). (**B**) Positions of LRET probes highlighted on the *P. furiosus* Rad50 AMPPNP-bound dimer (*Williams et al., 2011*). Gray lines show L51 of Rad50b interacting with each of the probe residues of Rad50a. (**C**) Plot of representative LRET emission decays versus time after Tb[3+]-chelate donor excitation. (**D**) Table of the LRET-determined distances (in Å) for the identity pairs, where the same residue is labeled in each protomer, in MR[NBD] and full-length MR complexes. One Rad50 was labeled with either with Bodipy FL or Cy3 acceptor and the other with Tb[3+]-chelate donor. Green, orange, and purple shaded cells indicate distances in the 'closed,' 'partially open,' and 'open' conformations, respectively. Values are the mean and standard deviation of at least three replicates.

The online version of this article includes the following figure supplement(s) for figure 1:

**Figure supplement 1.** MR[NBD] complexes made with cysteine mutants of Rad50 are active.

**Figure supplement 2.** HADDOCK active and passive Mre11 to Rad50 ambiguous interaction restraints (AIRs) are shown on the closed HADDOCK model.

## Results

### Multiple LRET probe positions provide a network of measurements

LRET experiments employ a luminescent lanthanide donor and fluorophore acceptor pair with an appropriate Förster radius ($R_0$) (*Zoghbi and Altenberg, 2018*). Due to the long lifetime of the luminescence signal, LRET has several advantages over FRET. First, the lanthanide luminescence excited state lifetime is milliseconds, whereas the fluorescent acceptor lifetime is nanoseconds. This difference allows LRET to be measured after background fluorescence has decayed, eliminating artifacts

from direct acceptor excitation/emission and scattering by the sample. Second, LRET measurements are insensitive to incomplete labeling because millisecond timescale donor-sensitized acceptor signal only occurs when a donor and fluorophore are in proximity. LRET probes are introduced into a protein most easily through a thiol-maleimide reaction with a unique cysteine. For Pf Rad50$^{NBD}$, where the coiled-coil domains are truncated, single cysteines were introduced into the naturally cysteine-less construct, and MR$^{NBD}$ activity was tested to ensure an active complex (*Figure 1—figure supplement 1*). Mutations were made primarily in loop regions to minimize disruptions to the protein fold. In all, six separate single-cysteine mutations were made throughout Rad50$^{NBD}$ (*Figure 1B*). These cysteines were subsequently labeled with thiol-reactive LRET donor or acceptor molecules. To make the MR$^{NBD}$ complex for LRET experiments, equimolar amounts of donor-labeled (Tb$^{3+}$-chelate) Rad50$^{NBD}$ and acceptor-labeled (Bodipy FL or Cy3) Rad50$^{NBD}$ were mixed with twice the molar ratio of Mre11 so that 50% of the resulting M$_2$R$^{NBD}_2$ complexes had one donor and one acceptor fluorophore (on separate Rad50$^{NBD}$ protomers). Not only were identical cysteines mixed within a complex (e.g., Tb$^{3+}$-S13C and Bodipy-S13C), but complexes were also made where cysteine mutants were mixed with other cysteine mutants (e.g., Tb$^{3+}$-S13C and Bodipy-L51C). Additional distance measurements were obtained for a given LRET pair by changing the identity of the acceptor as the Förster radius for Tb$^{3+}$ and Cy3 (61.2 Å) is longer than that of Tb$^{3+}$ and Bodipy FL (44.9 Å). Finally, donor- and acceptor-labeled cysteines within mixed pairs were swapped (e.g., Tb$^{3+}$-S13C and Bodipy-L51C versus Tb$^{3+}$-L51C and Bodipy-S13C) for added confidence in measurements. In total, 20 different cysteine pairs resulted in 52 unique samples that gave 54 total measured distances between the two Rad50 protomers in the MR$^{NBD}$ complex (*Supplementary file 1*).

## LRET measurements reveal three distinct sets of distances

Following laser excitation of the Tb$^{3+}$-chelate moiety and a 200 μs delay, donor-sensitized Bodipy FL or Cy3 fluorescence emission decay curves were collected for each of the MR$^{NBD}$ LRET samples at 5°C in the presence of 5 mM Mg$^{2+}$ and 2 mM ATP (*Figure 1C*). Under these conditions, Rad50$^{NBD}$ should be >99% bound to ATP as the K$_D$ for ATP is ~3 μM and there is no measurable ATP hydrolysis in 1 hr at 50°C. Collecting LRET data at 50°C (the maximum temperature for the fluorimeter) approaches the physiological temperature range for the hyperthermophilic Pf MR (>60°C) while minimizing Rad50-catalyzed ATP hydrolysis. In multiexponential fits, the emission decays were best described by two or three exponentials depending on the identity of the LRET pair (see Materials and methods). In all cases, the first lifetime (<100 μs) is a function of instrument response time and was discarded (*Cooper and Altenberg, 2013*; *Zoghbi et al., 2017*; *Zoghbi et al., 2012*). The Tb$^{3+}$-chelate luminescence decays were also recorded at each probe position in donor-only-labeled MR$^{NBD}$ complexes. As expected, the value of the Tb$^{3+}$-chelate lifetime changed with the local environment of each cysteine. Using these Tb$^{3+}$-chelate donor lifetimes, combined with the donor-sensitized acceptor lifetimes and the R$_0$ of the dye pair in the sample, distances were calculated between probes for each LRET pair.

For the majority of the 20 cysteine pairs analyzed, combining the data for the Bodipy/Tb$^{3+}$- and Cy3/Tb$^{3+}$-labeled samples gave three distinct distances. For 10 of the pairs (e.g., L51C-L51C), the longer distance (d2$^{Bodipy}$) in the Bodipy/Tb$^{3+}$ samples matched the shorter distance (d1$^{Cy3}$) in the Cy3/Tb$^{3+}$ samples, and the Cy3/Tb$^{3+}$ samples gave a second, longer distance (d2$^{Cy3}$) (*Figure 1D*, *Supplementary file 1*). This longer distance became 'visible' in samples where Cy3 was the acceptor because the R$_0$ for Tb$^{3+}$ and Cy3 is longer. For four of the pairs (e.g., L51C-A66C), the one Bodipy/Tb$^{3+}$ distance did not overlap with the two Cy3/Tb$^{3+}$-determined distances, and the combined data resulted in three distances. For two pairs (A66C-A66C and S93C-S93C), only one distance was seen in the Bodipy/Tb$^{3+}$ data, while the Cy3/Tb$^{3+}$ data contained two distances. In these samples, the d$^{Bodipy}$ matched d1$^{Cy3}$ for a total of two distances. And finally, for four pairs (e.g., S13C-S93C) only Cy3/Tb$^{3+}$ samples were made, resulting in two distances. Together, these data illuminate the presence of a third solution state in addition to the closed and partially open conformations.

To confirm that the distances observed in MR$^{NBD}$ were the same in full-length MR, all of the cysteine mutations were also introduced into full-length Pf Rad50, which contains the long coiled-coil domains and apical zinc hook motif. We previously reported that the two native cysteines in the zinc hook motif of full-length Rad50 are not efficiently labeled by the LRET probes and do not result in LRET donor-sensitized acceptor signal (*Boswell et al., 2020*). Tb$^{3+}$-chelate donor-only lifetimes measured in these mutants were identical to those measured for the Rad50$^{NBD}$ construct, indicating that the local

environments of the introduced cysteines do not change between full-length and NBD constructs. Unfortunately, because full-length Rad50 dimerizes at the zinc hook, cysteine mutants could not be mixed and only 'identity' LRET pairs (e.g., Bodipy-L51C and Tb$^{3+}$-L51C) could be made. Nonetheless, for all LRET probe positions measured, the distances between full-length MR cysteine pairs were within a few Ångströms of those measured in MR$^{NBD}$ (*Figure 1D*).

## HADDOCK models of three MR$^{NBD}$ conformations

The measured LRET distances were input as unambiguous restraints in the HADDOCK molecular docking program (*Dominguez et al., 2003*; *van Zundert et al., 2016*), defining the Cβ-Cβ distance between the LRET-labeled residues. The unambiguous restraints included symmetrical distances for each LRET pair (e.g., both Rad50a protomer L51 to Rad50b protomer S13 and Rad50a S13 to Rad50b L51). Given the relatively large size and unknown orientation of the fluorophores and their linkers with respect to the protein, ±5 or ±7 Å bounds (for distances less than and greater than 75 Å, respectively) were used for the unambiguous LRET distance restraints in HADDOCK. These unambiguous restraints were used to dock two Rad50$^{NBD}$ protomers with one Mre11 dimer in a three-body docking simulation.

The 'closed' HADDOCK model fit very closely to the measured LRET distances (*Figure 2A*, *Figure 2—video 1*, *Supplementary file 2*). HADDOCK returned 190 structures in five clusters, with 164 in the top-scored cluster. The Rad50 dimer formed in this model is nearly identical to the AMPPNP-bound dimer structure of Pf Rad50$^{NBD}$ (PDB: 3QKU, all-atom root-mean-square deviation [RMSD] = 1.11 Å) (*Williams et al., 2011*). Except in two cases, the Cβ-Cβ distances between the Rad50 LRET pairs in the HADDOCK model were within ±5 Å of their respective unambiguous LRET restraint. LRET pairs A66-A66 and A66-S93 deviated more significantly with differences of 6.6 and 11.0 Å, respectively. This deviation could arise from slight differences in the loop structures between the solution (LRET) and crystal states (i.e., input PDB; note that HADDOCK does not move backbone atom positions during model refinement). Specifically, we observed the expected relative positions within the associated Rad50s for residues in the Walker A motif (N32) of one protomer and the D-loop (D829) and signature motif (S793) of the other (*Figure 2D*, top, *Figure 2—video 1*). In this closed model, Rad50 interacts with Mre11 via the capping domain and along the top of the nuclease domain as observed in the *Thermotoga maritima* (PDB: 3THO; *Möckel et al., 2012*), *Methanocaldococcus jannaschii* (PDB: 3AV0; *Lim et al., 2011*), and *E. coli* (PDB: 6S6V; *Käshammer et al., 2019*) nucleotide-bound structures (with Cα RMSDs of 4.9 Å, 1.9 Å, and 4.7 Å, respectively) (*Figure 2—figure supplement 1A*). Like the *M. jannaschii* structure, each Rad50 protomer makes contact with only one of the Mre11 capping domains mainly through interactions between capping domain β18 and Rad50 Lobe II αE and β8–10. In total, eight and four ionic or hydrogen bond interactions are made between Rad50 and the Mre11 capping and nuclease domains, respectively. In particular, unique contacts not seen in the other two conformations are made between Rad50 E831 and Mre11 H17 in the nuclease domain and Rad50 E758/E761 and Mre11 Y325/K327 in the capping domain. The combination of these interactions occludes the Mre11 nuclease active site for dsDNA, as previously described (*Lim et al., 2011*; *Möckel et al., 2012*).

The 'partially open' HADDOCK simulation returned nine clusters of models, and analysis of the top four clusters concluded that the distances between LRET pairs for each cluster had an average standard deviation of ~0.3 Å when compared among the top three clusters and increased to ~0.7 Å when adding the fourth (*Figure 2B*, *Figure 2—video 1*, *Supplementary file 2*). 13 out of 20 of the Cβ-Cβ distances in the HADDOCK model are within ±5.7 Å of the measured LRET distances (*Supplementary file 2*). Interestingly, all of the pairs with larger (>5.7 Å) deviations included either A66 or S93, again suggesting that the position of those loops differs between the crystal and solution conditions. In the partially open model, only 4.2 Å separates the Rad50 protomers at their closest point between the two D829 residues (Cα-Cα distance). In this conformer, the Walker A/D-loop interactions are no longer formed between the Rad50 protomers as the Walker A motifs have rotated out of the Rad50-Rad50 interface (*Figure 2D*, bottom, *Figure 2—video 1*). Instead, the D-loop and signature motifs face one another and are stacked in the interface. Although there are still significant interactions between the Rad50 protomers and the Mre11 capping and nuclease domains, several of which are maintained from the closed conformation, the partially open conformation has rotated by 22° to interact differently with the capping domain compared to the closed structure (*Figure 2—figure supplement 1B*). Specifically, interactions with the W308-D313 loop in the Mre11 capping domain are similar between

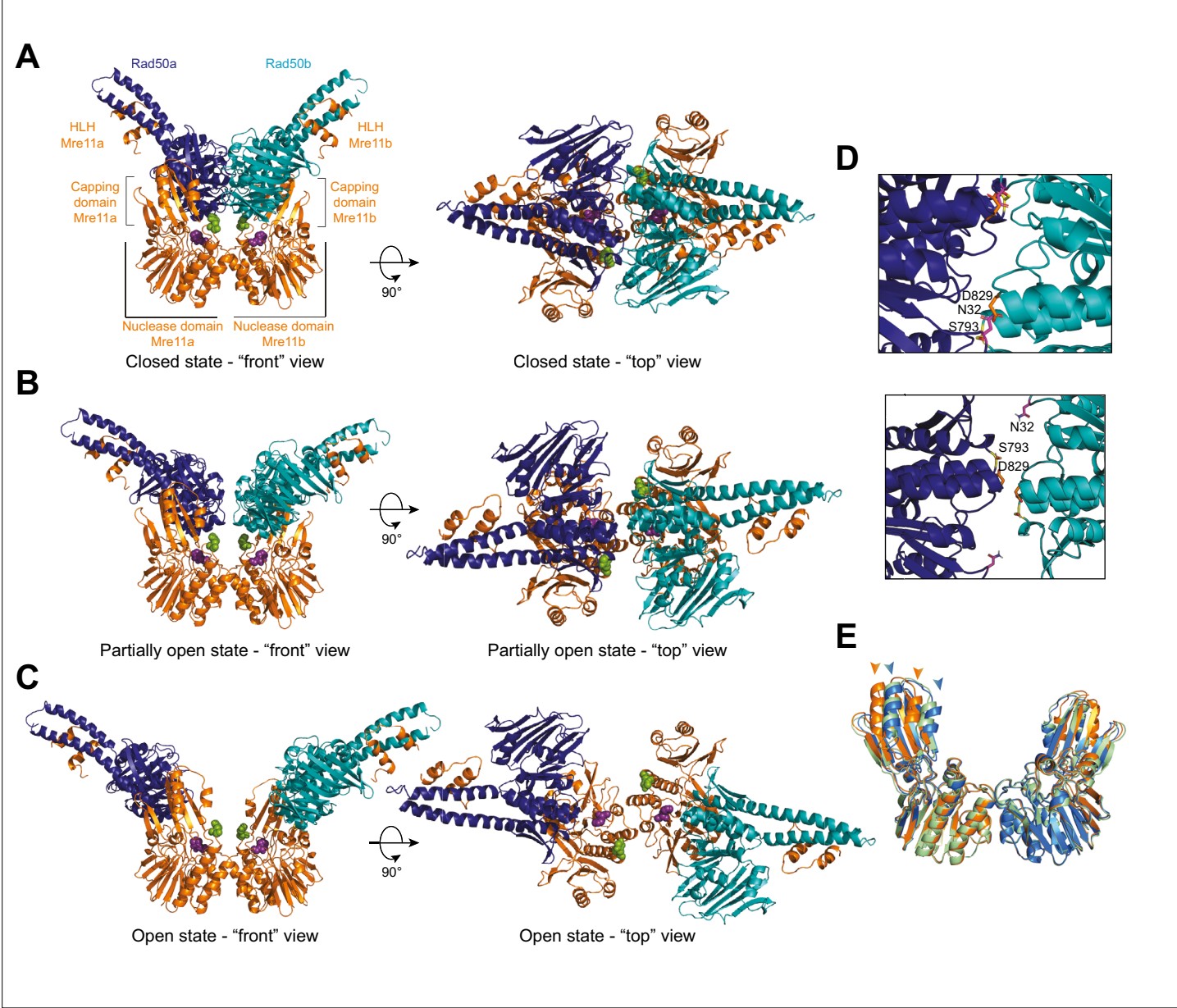

**Figure 2.** The Pf MR^NBD ATP-bound complex has at least three conformations in solution. HADDOCK structural models of the (**A**) closed, (**B**) partially open, and (**C**) open MR^NBD complex. Mre11 H85, which helps to coordinate the catalytic $Mn^{2+}$ ions, is shown as purple spheres, and Mre11 H17, the wedge residue, is shown as green spheres. (**D**) The Rad50-Rad50 interface with the Walker A residue (N32, magenta) from Rad50b and the Signature helix (S793, yellow) and D-loop (D829, orange) residues of Rad50a indicated in the closed (top) and partially open (bottom) conformations. (**E**) Overlay of the Mre11 dimers from the closed (orange), partially open (green), and open (blue) HADDOCK models showing that the capping domain moves out (arrows) to accommodate the associated Rad50s in the closed conformation.

The online version of this article includes the following video and figure supplement(s) for figure 2:

**Figure supplement 1.** Mre11 and Rad50 make different interactions in the three conformations of the MR^NBD complex.

**Figure supplement 2.** Data from one single luminescence resonance energy transfer (LRET) probe position do not dominate the HADDOCK structure calculations.

**Figure 2—video 1.** The movie depicts the transitions between the closed, partially open, and open conformations first shown from the 'side' view and then from the 'top' view.

https://elifesciences.org/articles/69579/figures#fig2video1

the two conformations, but the Rad50 interactions with β18 have been broken in partially open, and Rad50 is now interacting with Mre11 β16 and β17 residues instead. Moreover, the movement of Rad50 has allowed the Mre11 capping domains to rotate inward toward the nuclease domains (*Figure 2E*). We observe a total of nine ionic or hydrogen bond interactions between the Mre11 capping domain and Rad50 and five between the nuclease domain and Rad50. In the capping domain, unique interactions occur between Mre11 K277 and Rad50 E750 and Mre11 R303 and Rad50 E754. Even with these contacts between Rad50 and Mre11, DNA could access the nuclease active site of Mre11.

Finally, HADDOCK returned 14 clusters of models for the 'open' complex, and analysis of the top four clusters showed that the distances between LRET pairs for each cluster had an average standard deviation of ~0.7 Å among the top three clusters and ~1.2 Å when adding the fourth. 10 out of 14 of the Cβ-Cβ distances in the top cluster are within ±5.3 Å of the input LRET unambiguous restraints, whereas the remaining 4 are within ±9.3 Å (*Figure 2C*, *Figure 2—video 1*, *Supplementary file 2*). The Rad50 protomers have moved apart considerably (~41 Å D829-D829 Cα-Cα distance). Moreover, the orientation of the Rad50 protomers with respect to Mre11 has changed significantly (*Figure 2—figure supplement 1C*) having rotated 13° and the base of the coiled-coil translated ~23 Å away from the partially open model. Because of this rotation and translation, there are minimal contacts (only two) with the Mre11 capping domain, which now occur between Rad50 β10 and Mre11 β16 and the C-terminus of this construct. Now, six ionic interactions are formed between the Mre11 nuclease domain and Rad50. Within the nuclease domain, unique interactions occur between Mre11 R177 and E181 on helix αE with Rad50 E841 and R842 and between Mre11 E152 on helix αD and Rad50 K860. In the capping domain, Mre11 K279 makes a unique interaction with Rad50 E783. With Rad50 rotated fully away, the capping domains move even closer to the nuclease domain and both Mre11 nuclease active sites are now fully accessible to dsDNA for exonuclease activity.

To ensure that the unambiguous distance restraints obtained from one probe position were not dominating the structure calculations, HADDOCK runs were performed with systematic dropouts of all restraints calculated from a specific cysteine position. For example, for the L51C probe position, L51C-S13C, L51C-L51C, L51C-A66C, L51C-S93C, L51C-N774C, and L51C-V866C distances were all removed from the unambiguous restraints, and HADDOCK runs were repeated for each of the three sets of distances (closed, partially open, and open). In general, none of the dropouts appreciably changed the overall conformation of any of the states (*Figure 2—figure supplement 2*).

## Destabilizing solution-state conformers alters MR activity

Next, to decrease the stability of one or two conformations over the others, charge reversal mutations were made to several Mre11 residues that directly interact with Rad50. As there are a handful of shared interactions between Mre11 and Rad50 in the various conformation combinations, it was impossible to completely disrupt a given state. The Mre11 mutants were combined with full-length Rad50 to make MR complex, and then both Mre11 nuclease activity and Rad50 ATP hydrolysis activity were tested (*Figure 3B–E*). For Mre11 nuclease activity, three sets of fluorescence-based experiments were performed in 384-well plates (*Figure 3B–D*). The Exo2 assay interrogates the 3′-to-5′ $Mn^{2+}$-dependent exonuclease activity as only 2 base-pairs need to be excised before the fluorescent 2-aminopurine nucleotide is released from the 3′-end of the double-stranded duplex (*Figure 3B*). The Exo11 signal results from a combination of exonuclease and endonuclease activity as the fluorescent nucleotide is 11 base-pairs from the 3′ end of the dsDNA and requires either sequential nuclease functions (e.g., endonuclease followed by exonuclease) or processive exonuclease activity to free it from the duplex (*Figure 3C*). Finally, the Cy3/BHQ2 ssDNA assay monitors endonuclease activity since the ssDNA needs to be cleaved to separate the quencher from the fluorophore (*Figure 3D*). In addition to these plate-based fluorescent assays, the cleavage of four different DNA substrates was monitored on 15% denaturing polyacrylamide gels: a 40-nucleotide ssDNA, a 40-nucleotide dsDNA, a 36-nucleotide hairpin DNA construct that includes a 2-nucleotide 3′-overhang, and a 50-nucleotide dsDNA with five phosphorothioate bonds that block 3′-to-5′ exonuclease activity at the 3′ end of the labeled strand (*Figure 3—figure supplement 1*). MR complexes made with Pf Mre11 H52S (exonuclease inactive) and H85S (exo- and endonuclease inactive) mutants were used as controls in all of these assays (*Hopfner et al., 2001*; *Williams et al., 2008*). In total, when the results of these various experiments were compared, we found that the Exo2 data mirrored the results of the 40-mer dsDNA cleavage in the gels in the absence of ATP – the 3′ nucleotide of the dsDNA was removed, but no

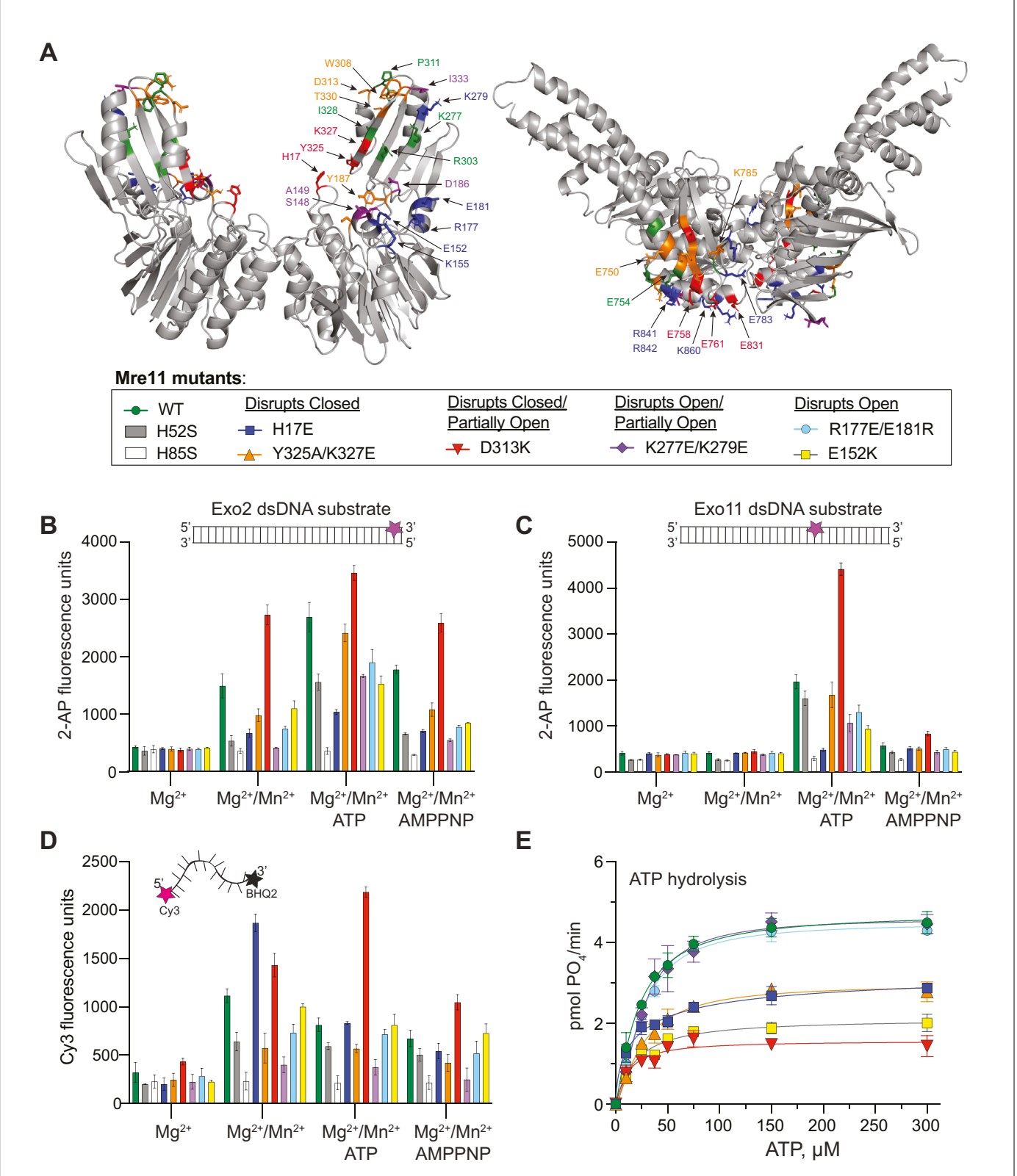

**Figure 3.** Partially open and open conformations of the MR complex are important for nuclease activities. (**A**) Mre11 (left) and Rad50 (right) dimers showing residues involved in protein-protein interactions only in closed (red), only in partially open (green), and only in open (blue) conformations or common to closed and partially open (orange) or common to all three (purple). The box below is a legend providing the color of each mutant for the activity data shown in (**B–E**) as well as the conformation the mutant was designed to destabilize. (**B, C**) Full-length MR complex nuclease activity on

*Figure 3 continued on next page*

*Figure 3 continued*

the Exo2 (**B**) or Exo11 (**C**) dsDNA substrates. Position of fluorescent 2-AP is indicated with a star on the cartoon of each substrate. (**D**) Full-length MR complex endonuclease activity on the Cy3/BHQ2-labeled ssDNA substrate. (**E**) ATP hydrolysis activity for full-length MR complexes containing the indicated Mre11 mutants. Data are the mean and standard deviation of n ≥ 3 replicates.

The online version of this article includes the following figure supplement(s) for figure 3:

**Figure supplement 1.** Nuclease activities of the full-length MR complex resolved by denaturing polyacrylamide gel electrophoresis.

**Figure supplement 2.** Unique interactions are made between Mre11 and Rad50 in the three conformations of MR$^{NBD}$.

---

further cleavage products were detected (*Figure 3B*, *Figure 3—figure supplement 1A*). The Exo11 results, on the other hand, very closely resembled those of the dsDNA 40-mer and hairpin DNA gels in the presence of ATP (*Figure 3*, *Figure 3—figure supplement 1B C*). Interestingly, the cleavage pattern of the blunt-ended dsDNA, but not the hairpin with the 3′ overhang, showed intense bands where the 3′ nucleotide had been cleaved off, and both of these substrates had a 'ladder' of cleavage products. Thus, it appears that the 3′-overhang of the hairpin dsDNA either precludes the 3′-to-5′ exonuclease activity of the complex or MR exonuclease activity occurs at the fifth nucleotide from the end. Unexpectedly, whereas the nuclease-inactive H85S showed no activity on any of the four gel-based DNA substrates, the previously described exonuclease-impaired H52S did show activity on all of these substrates. Note, to the best of our knowledge, that there are no reports in the literature of the nuclease activity of H52S on any substrate other than the Exo2 in the absence of ATP; therefore, the cleavage products observed with that mutant were surprising and perhaps further demonstrate the importance of endonuclease activity in Pf MR (*Figure 3—figure supplement 1*). Finally, the Cy3/BHQ2 ssDNA results were nearly identical to the results for the cleavage of the 40-nucleotide ssDNA in the gel as well as to the 3′-end phosphorothioate-blocked dsDNA (*Figure 3D*, *Figure 3—figure supplement 1D E*). Although both of these were worse substrates for MR as compared to the 40-mer dsDNA and the hairpin, they were both cleaved in the absence of ATP. Below, to describe the effects that each destabilizing Mre11 mutant has on MR nuclease activity, we used the results of the quantitative Exo2, Exo11, and Cy3/BHQ2 fluorescence plate assays (*Figure 3*).

In the closed conformation, Mre11 nuclease domain residue H17 interacts with Rad50 E831 (*Figure 3—figure supplement 2A*). MR H17E had ~30% of wild-type exonuclease activity on Exo2 DNA but does not cleave Exo11 DNA (*Figure 3B and C*). *Williams et al., 2008* identified H17 as a 'wedge residue' that helps to unwind the dsDNA helix. H17E is, however, a competent ssDNA endonuclease, showing nearly twice the activity of WT MR on the Cy3/BHQ2 ssDNA substrate in the absence of ATP (*Figure 3D*). This result is consistent with previous reports that dsDNA and ssDNA substrates use different binding sites on Mre11 (*Rahman et al., 2021*; *Rahman et al., 2020*). When ATP is added, the ssDNA endonuclease activity of MR H17E decreases to WT levels. Although MR H17E cannot cleave dsDNA, it showed robust activity when the 3′-end of the DNA duplex is blocked with phosphorothioate bonds (*Figure 3—figure supplement 1E*). Moreover, the banding pattern of the 3′-phosphorothioate-blocked substrate was different for MR H17E when compared to wild-type or the other mutants. Therefore, we suggest that the cleavage products observed on this modified dsDNA for the H17E mutant were all from endonuclease activity, whereas the products of the other MR complexes originate from a combination of endo- and exonuclease activities. Finally, MR H17E decreased the V$_{max}$ of ATP hydrolysis to ~60% of wild-type MR (*Figure 3E*), confirming that this residue assists in destabilizing the closed conformation from which hydrolysis proceeds.

Mre11 Y325 and K327 are located in the capping domain on β18 and interact with E761 and E758 of Rad50, respectively, in the closed conformation (*Figure 3—figure supplement 2A*). In the presence of ATP, the Y325A/K327E double mutant had ~85% of the exonuclease activity of wild-type MR, but only ~50% of the activity for the Exo2 substrate without ATP (*Figure 3B and C*). MR Y325A/K327E had an approximately twofold decrease in endonuclease activity on the Cy3/BHQ2 ssDNA substrate (*Figure 3D*). Additionally, MR Y325A/K327E decreased the V$_{max}$ of ATP hydrolysis to the same level as MR H17E (*Figure 3E*). Thus, like MR H17E, destabilizing the closed conformation through the Mre11 Y325A/K327E mutant had the predictable effect of decreasing Rad50 ATP hydrolysis.

Mre11 D313 contacts Rad50 K785 in both the closed and partially open conformations (*Figure 3—figure supplement 2A B*). D313 and its neighboring residues in a loop at the top of the capping domain might be acting as a pivot point for Rad50 to rotate between the two states. Surprisingly, MR

D313K had significantly increased nuclease activity on all of the substrates (*Figure 3*, *Figure 3—figure supplement 1*). In fact, D313K showed shorter cleavage products, indicating more nuclease cleavage, in the gel-based assays when compared to wild-type or the other mutants (*Figure 3—figure supplement 1A–C*). An approximately twofold increase in activity was observed for Exo2 in the absence of ATP (*Figure 3B*); thus, we hypothesize that destabilizing both the closed and partially open conformations increases the population of the open conformation that accommodates dsDNA substrate. Activity against the Exo11 substrate increased ~2.5-fold in this mutant (*Figure 3C*), and the Cy3/BHQ2 ssDNA results showed an ~1.3- and 3-fold increase in cleavage over WT in the absence and presence of ATP, respectively (*Figure 3D*). In contrast to its high levels of nuclease activity, MR D313K reduced the $V_{max}$ for ATP hydrolysis by more than 50%, which was expected since the closed state is destabilized (*Figure 3E*). Therefore, Mre11 nuclease and Rad50 ATPase results for the D313K mutant demonstrate that the ATP hydrolysis function is not directly correlated with nuclease activity.

In the partially open complex, Mre11 K277 in β16 interacts with Rad50 E750, which is in αE at the base of a coiled-coil, whereas in the open complex, Mre11 K279 also in β16 interacts with Rad50 E783 in the short loop between β9 and β10 (*Figure 3—figure supplement 2B C*). As these two residues are close in sequence space, a double mutant was constructed to destabilize both partially open and open conformations. The most striking feature of MR K277E/K279E was that it had no nuclease activity on Exo2 without ATP but increased to ~50% of wild-type levels when ATP was added (*Figure 3B*). No other mutant tested required ATP for the Exo2 substrate, implying that it requires hydrolysis to open the complex before dsDNA can bind. Moreover, this mutant displayed the lowest activity in the presence of AMPPNP, suggesting that the combination of the mutations and non-hydrolyzable analog effectively stabilized the closed state, fully occluding the dsDNA substrate. On the Exo11 substrate, the activity was also ~50% of wild-type (*Figure 3C*). On the Cy3/BHQ2 ssDNA, the endonuclease activity decreased to ~20% of wild-type (*Figure 3D*). In fact, MR K277E/K279E had the worst endonuclease activity of all of the mutants tested, except for the nuclease-deficient H85S, suggesting that the closed state is not productive for endonuclease activity. This double mutant has no effect on ATP hydrolysis activity (*Figure 3E*) since the closed conformation can readily form.

Finally, mutants were made for Mre11 R177, E181, and E152, which all make unique contacts in the open conformation (*Figure 3—figure supplement 2C*). These residues are along the 'top edge' of the Mre11 nuclease domain in αD and αE, and R177/E181 and E152 are structurally homologous to part of the so-called 'latching loop' and 'fastener,' respectively, of the *E. coli* SbcCD cryo-EM 'cutting state' structure (*Käshammer et al., 2019*). The MR R177E/E181R double mutant had ~60% of wild-type nuclease activity (*Figure 3B and C*) and 60% (no ATP) to 80% (with ATP) Cy3/BHQ2 ssDNA endonuclease activity (*Figure 3D*), but wild-type ATP hydrolysis activity (*Figure 3E*), which is consistent with there being less open and more closed conformation. On the other hand, MR E152K, which interacts with Rad50 K860 in αE, showed more impaired nuclease activity than R177/E181, but nearly WT endonuclease activity, and, surprisingly, decreased the $V_{max}$ for ATP hydrolysis by more than 50% (*Figure 3*). As this mutant should readily form the closed conformation, this result was unexpected and, like D313K, demonstrates the independence of nuclease activity on ATP hydrolysis. Similar to the K227E/K279E mutant, both MR R177E/E181R and MR E152K had very little Exo2 activity in the presence of AMPPNP, again suggesting a complete stabilization of the closed conformation.

## Multiple conformations of MR persist for various substrate-bound states

The LRET distances used for the HADDOCK structural model calculations above were determined under saturating ATP conditions. Next, LRET measurements were collected for a subset of the LRET pairs without any nucleotide or in the presence of the non-hydrolyzable ATP analog ATPγS. LRET pairs S13-S13, L51-L51, and N774-N774 in both the MR<sup>NBD</sup> and full-length MR constructs were examined. These three residues are distributed across the dimerization face of Rad50 (*Figure 1B*), and together should illuminate any significant changes in distance between the two Rad50 protomers of the MR complex. MR<sup>NBD</sup> S13-L51 and N774-L51 LRET pairs were also examined. As reported in *Table 1*, three distances were observed for each LRET pair in the absence of nucleotide ('apo'), and these distances did not differ substantially from the distances observed under saturating ATP conditions. Although the 'open' distances all increased, most were within the error of the distance observed with ATP and all were within the ±7 Å of the HADDOCK inputs. When the non-hydrolyzable ATP analog ATPγS was

**Table 1.** Multiple conformations of the MR complex in various nucleotide- and DNA-bound states. Table of the luminescence resonance energy transfer (LRET)-determined distances (in Å) for a subset of LRET pairs in MR$^{NBD}$ or full-length MR determined in the absence of nucleotide (apo), ATP-bound, ATP γ S-bound, ATP- and hairpin DNA-bound, and ATP- and ssDNA-bound. For each LRET pair, the first Rad50 was labeled with either with Bodipy FL or Cy3 acceptor and the second Rad50 with Tb$^{3+}$-chelate donor. Green, pink, and purple shaded cells indicate distances in the 'closed,' 'partially open,' and 'open' conformations, respectively. For the distances describing the partially open state, the individual d2$^{Bodipy}$ and d1$^{Cy3}$ distances are both given. Values are the mean and standard deviation of at least three replicates.

| MR$^{NBD}$ LRET Pair | Condition | LRET closed (Bo) | LRET partially open (Bo, Cy3) | LRET open (Cy3) |
|---|---|---|---|---|
| | apo | 37.5 ± 0.7 | 46.2 ± 1.3, 46.1 ± 1.7 | 81.4 ± 2.2 |
| | ATP | 37.7 ± 0.6 | 45.4 ± 1.1, 47.0 ± 1.0 | 78.0 ± 2.0 |
| | ATPγS | 37.9 ± 0.1 | 47.7 ± 0.4, 48.0 ± 0.5 | 79.1 ± 1.1 |
| | ATP + Hairpin | 37.7 ± 1.2 | 47.9 ± 1.4, 47.2 ± 1.2 | 79.1 ± 1.0 |
| L51/Tb$^{3+}$- L51 MR$^{NBD}$ | ATP + ssDNA | 36.7 ± 0.8 | 46.8 ± 1.3, 45.4 ± 1.4 | 74.5 ± 0.7 |
| | apo | 33.7 ± 0.6 | 56.0 ± 0.3, 50.6 ± 1.5 | 85.0 ± 1.2 |
| | ATP | 35.7 ± 0.2 | 51.5 ± 2.3, 45.8 ± 1.8 | 83.5 ± 2.2 |
| | ATPγS | 35.6 ± 0.1 | 53.0 ± 1.5, 48.6 ± 0.2 | 82.6 ± 3.0 |
| | ATP + Hairpin | 35.4 ± 0.3 | 53.6 ± 0.5, 46.9 ± 1.3 | 82.7 ± 2.0 |
| N774/Tb$^{3+}$-N774 MR$^{NBD}$ | ATP + ssDNA | 35.1 ± 0.5 | 52.8 ± 0.4, 50.0 ± 2.7 | 75.3 ± 0.6 |
| | apo | 36.7 ± 1.0 | 55.6 ± 0.3, 46.0 ± 0.9 | 79.9 ± 2.2 |
| | ATP | 35.4 ± 0.3 | 51.1 ± 1.0, 50.9 ± 1.1 | 77.8 ± 1.7 |
| | ATPγS | 37.1 ± 0.1 | 51.9 ± 0.1, 47.0 ± 1.2 | 77.1 ± 0.2 |
| | ATP + Hairpin | 34.4 ± 1.2 | 54.9 ± 0.8, 47.8 ± 0.7 | 74.9 ± 0.2 |
| S13/Tb$^{3+}$-S13 MR$^{NBD}$ | ATP + ssDNA | 34.2 ± 0.7 | 53.9 ± 0.7, 48.0 ± 1.7 | 75.0 ± 0.8 |
| | apo | 34.7 ± 0.1 | 55.4 ± 1.0, 50.1 ± 0.8 | 84.1 ± 1.6 |
| | ATP | 36.7 ± 1.1 | 51.4 ± 0.9, 51.6 ± 0.3 | 79.7 ± 3.0 |
| | ATPγS | 36.0 ± 0.2 | 52.8 ± 0.2, 51.3 ± 0.2 | 79.7 ± 1.1 |
| | ATP + Hairpin | 34.7 ± 0.7 | 52.3 ± 0.4, 50.9 ± 0.1 | 77.2 ± 0.5 |
| S13/Tb$^{3+}$- L51 MR$^{NBD}$ | ATP + ssDNA | 35.1 ± 0.8 | 50.0 ± 0.5, 51.1 ± 0.4 | 74.5 ± 0.7 |
| | apo | 33.0 ± 0.8 | 53.0 ± 0.8, 48.9 ± 1.8 | 84.0 ± 0.9 |
| | ATP | 30.9 ± 0.2 | 51.5 ± 0.4, 49.6 ± 1.5 | 80.8 ± 1.9 |
| | ATPγS | 31.2 ± 0.3 | 49.6 ± 0.6, 47.6 ± 0.9 | 77.8 ± 0.2 |
| | ATP + Hairpin | 30.6 ± 0.3 | 49.9 ± 0.9, 51.1 ± 3.0 | 80.9 ± 0.9 |
| N774/Tb$^{3+}$-L51 MR$^{NBD}$ | ATP + ssDNA | 30.4 ± 0.4 | 49.6 ± 0.6, 50.7 ± 2.1 | 75.9 ± 1.2 |
| L51/Tb$^{3+}$-L51 full-length MR | apo | 35.4 ± 1.8 | 48.7 ± 0.8, 46.7 ± 0.6 | 75.7 ± 1.2 |
| | ATP | 37.3 ± 0.1 | 49.6 ± 0.2, 46.3 ± 0.1 | 74.7 ± 0.5 |
| | ATPγS | 35.3 ± 0.9 | 48.7 ± 0.5, 46.7 ± 0.4 | 74.4 ± 1.6 |
| | ATP + Hairpin | 34.8 ± 1.1 | 47.7 ± 0.3, 45.5 ± 0.4 | 72.6 ± 1.1 |
| | ATP + ssDNA | 34.9 ± 1.1 | 47.2 ± 0.4, 45.5 ± 0.6 | 71.0 ± 0.5 |

*Table 1 continued on next page*

*Table 1 continued*

| MR$^{NBD}$LRET Pair | Condition | LRET closed | LRET partially open | LRET open |
|---|---|---|---|---|
| N774/Tb$^{3+}$-N774 full-length MR | apo | 32.8 ± 0.6 | 46.8 ± 0.6, 48.3 ± 0.8 | 89.0 ± 3.1 |
| | ATP | 35.6 ± 0.3 | 50.8 ± 0.7, 50.5 ± 0.4 | 87.8 ± 0.7 |
| | ATPγS | 33.8 ± 0.3 | 46.7 ± 0.2, 50.1 ± 0.4 | 86.0 ± 1.5 |
| | ATP + Hairpin | 33.6 ± 0.3 | 45.5 ± 0.6, 46.7 ± 1.8 | 79.9 ± 1.8 |
| | ATP + ssDNA | 33.6 ± 0.2 | 45.5 ± 0.5, 46.7 ± 1.5 | 75.7 ± 0.8 |
| S13/Tb$^{3+}$-S13 full-length MR | apo | 36.8 ± 0.3 | 51.0 ± 1.0, 49.6 ± 0.4 | 77.9 ± 0.6 |
| | ATPγS | 35.8 ± 0.3 | 50.8 ± 0.2, 48.6 ± 0.7 | 76.3 ± 1.0 |
| | ATP | 36.4 ± 0.3 | 50.9 ± 0.2, 50.7 ± 0.2 | 79.4 ± 0.1 |
| | ATP + Hairpin | 36.4 ± 0.6 | 51.0 ± 0.6, 50.4 ± 0.7 | 75.1 ± 2.1 |
| | ATP + ssDNA | 36.9 ± 0.7 | 51.3 ± 1.0 49.7 ± 0.2 | 75.1 ± 0.2 |

added, three distances were again observed for each LRET pair, these distances did not vary more than 3 Å from the distances recorded in the ATP samples, and nearly all the distances were within the error of the measurement made with ATP. Thus, any changes in MR structure due to binding or hydrolyzing ATP are limited to local conformations invisible to LRET as we have employed it. Additionally, the LRET data on the ATPγS-bound MR demonstrate that the partially open state is a stable conformation of the MR complex and does not appear to be an intermediate in the Rad50 ATP hydrolysis catalytic cycle.

To examine whether or not the addition of DNA substrates affected any of the MR conformations, either a hairpin DNA, with 15 base-pairs and a 2-nucleotide 3′-end overhang, or a 36-nucleotide ssDNA were added to the LRET samples containing 2 mM ATP. The only consistent difference in LRET distances was observed in the ssDNA samples: the open distances were shorter than what was noted in all other conditions (*Table 1*), indicating that the two Rad50s in the complex may be slightly closer together. None of the changes in LRET distances in the apo, ATPγS, or DNA-bound samples were large enough to necessitate recalculating HADDOCK. Note that our LRET measurements would not be able to detect if the Rad50 protomers were moving relative to the Mre11 dimer, as observed by cryo-EM in the *E. coli* SbcCD (MR homolog) 'cutting' state (*Käshammer et al., 2019*), or if the Mre11 protomers were moving relative to each other (*Käshammer et al., 2019*; *Saathoff et al., 2018*). Unfortunately, it is likely impossible to measure LRET between Mre11 and Rad50 within an MR complex because of cross-talk between the different protomers (i.e., Rad50a to Mre11a, Rad50b to Mre11b, Rad50a to Mre11b, and Rad50b to Mre11a) within the three states.

The different nucleotide-bound states and DNA substrates are most likely altering the population distribution of the various conformations of the MR complex. Unfortunately, it is not feasible to assign relative populations to these structures as we may not have a complete inventory of states, we do not know to what extent other structures (e.g., extended or 'cutting' state) are populated, and each LRET decay curve only provides population information for the one or two conformations that are described by the data fitting.

## SAXS corroborates HADDOCK MR$^{NBD}$ models

To further validate the multiple conformations indicated by the LRET data, SAXS data was collected for samples of MR$^{NBD}$ with and without ATP. The SAXS profiles for wild-type MR$^{NBD}$ were compared to our three models using the *FoXS* webserver (*Schneidman-Duhovny et al., 2016*; *Schneidman-Duhovny et al., 2013*), which calculates the SAXS profile from an input structural model and compares that to an experimental SAXS profile. The ATP-bound MR$^{NBD}$ experimental SAXS profile fit best to the partially open model ($\chi^2$ = 0.34) followed by the closed model ($\chi^2$ = 0.51), based on the single-state $\chi^2$ values (*Figure 4*). The open model did not fit well to the experimental SAXS data ($\chi^2$ = 13.6). Conversely, for the ATP-free MR$^{NBD}$ SAXS profile, a reasonable fit was only obtained with the open model ($\chi^2$ = 1.26), although a discrepancy between the experimental and back-calculated scattering data was observed for scattering angles of ~0.1–0.2 Å$^{-1}$ (*Figure 4A*, left). Next, *MultiFoXS* was

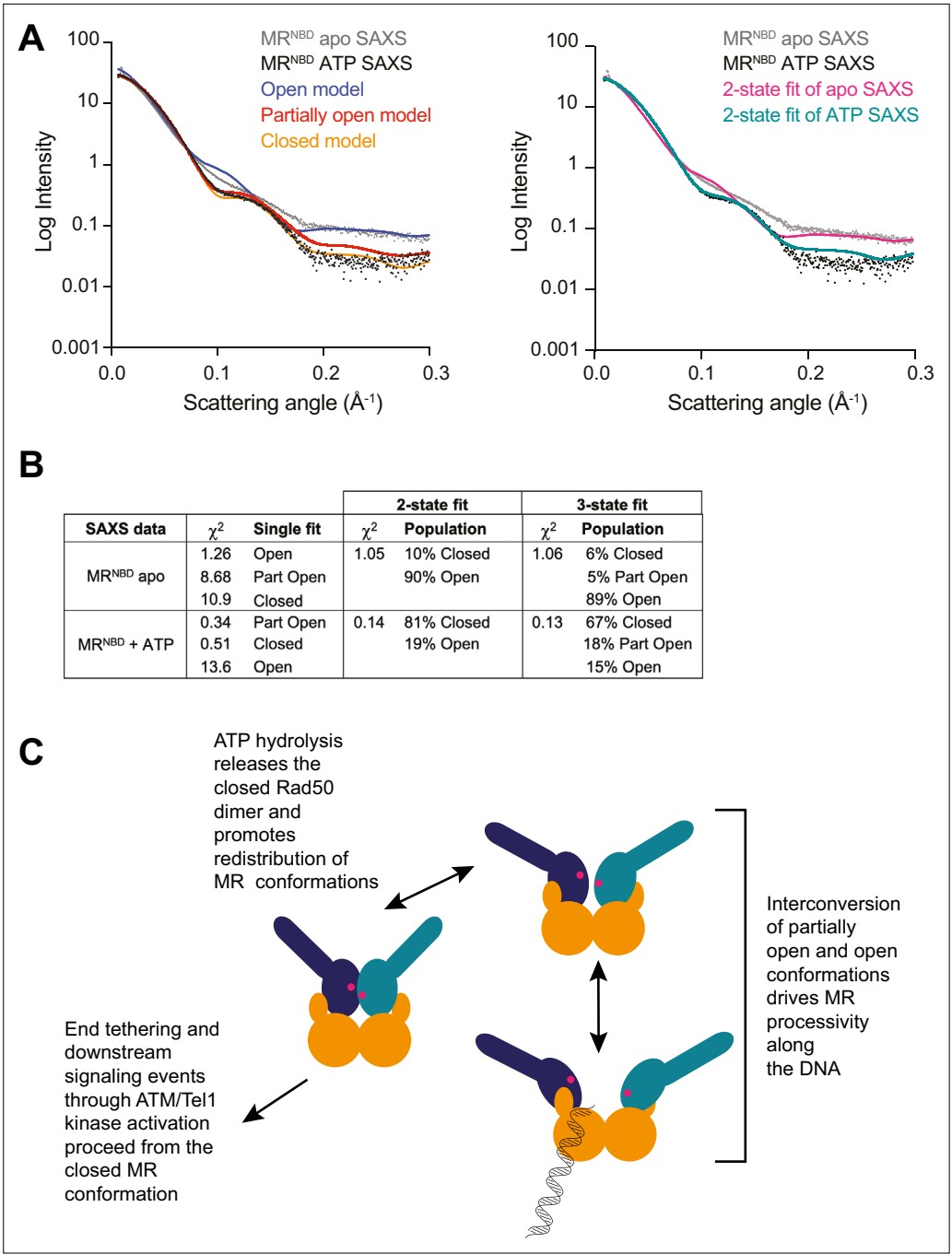

**Figure 4.** All three MR conformations play a role in the function of the complex. (**A**) The left panel shows experimental small-angle X-ray scattering (SAXS) data for apo (gray) and ATP-bound (black) MR$^{NBD}$ superimposed with *FoXS*-calculated theoretical SAXS curves for the closed (orange), partially open (red), and open (blue) HADDOCK models. The right panel shows the experimental SAXS data for apo (gray) and ATP-bound (black) MR$^{NBD}$ superimposed with *MultiFoXS*-calculated theoretical SAXS curves for the two-state fits of the HADDOCK models, given in (**B**), for the apo (magenta) and the ATP-bound (teal) data. (**B**) Goodness of fits of experimental to theoretical SAXS curves calculated by *FoXS* and *MultiFoXS*. For fits to two or three of the models, the populations from the fit are shown. (**C**) Proposed model of the functional role of conformational heterogeneity in the MR complex.

The online version of this article includes the following figure supplement(s) for figure 4:

**Figure supplement 1.** Rosetta-refined models of the MR$^{NBD}$ conformations.

used to fit the apo and ATP-bound MR$^{NBD}$ SAXS profiles to either two or three of the LRET-derived models simultaneously (*Figure 4A*, right, and B). The fit of the ATP-bound SAXS profile to two models improved the $\chi^2$ significantly and combined populations of open (19%) and closed (81%) conformations (*Figure 4B*). The fit using all three models did not improve the $\chi^2$ but did assign populations of 15% open, 18% partially open, and 67% closed. The two-state fit to the apo MR$^{NBD}$ SAXS profile only slightly improved the $\chi^2$ (1.26 vs. 1.05) and gave 90% open and 10% closed, while the three-state fit gave 89% open, 5% partially open, and 6% closed again without improving $\chi^2$ further. Here, minor improvements in the fit were observed in the 0.1–0.2 Å$^{-1}$ range; however, the quality was still not as good as the multistate fits to the ATP-bound SAXS data, implying that an additional, unobserved state might be populated in apo MR$^{NBD}$. Thus, the SAXS data generally support the presence of multiple states as observed in LRET data and demonstrate the expected shift in population to the closed form in the presence of ATP.

## Discussion

The disulfide bonds and non-hydrolyzable ATP analogs used in all of the closed MR$^{NBD}$ crystal structures and the selection for closed particles during the *E. coli* SbcCD cryo-EM model refinement imply that there is more than one conformation of ATP-bound MR. Nevertheless, the ATP-bound closed form has been referred to as the 'resting' state given the cellular concentration of ATP and the $K_D$ for ATP-binding to Rad50 (*Käshammer et al., 2019*). Even so, both closed and partially open conformations were also identified in apo MR (*Boswell et al., 2020*). In fact, the existing structural data (*Deshpande et al., 2014*; *Williams et al., 2011*) and the LRET-derived HADDOCK models presented here provide evidence for at least three conformations for Pf MR$^{NBD}$ in solution: closed, partially open, and open. Our data demonstrate that the MR complex samples these global conformations in both the ATP-free and ATP-bound forms with nucleotide binding shifting their relative populations. It is possible that the partially open state is a 'transient' intermediate on the pathway between the closed and open or extended structures. However, LRET data collected in the presence of the non-hydrolyzable analog ATPγS also revealed the presence of the partially open state. Additionally, the LRET-derived distances used to define the partially open state were calculated from fluorescence lifetimes of ~0.5 ms. Thus, the partially open state must be populated for at least 5 ms to be observed by LRET, offering a lower limit for the lifetime of the partially open state.

As LRET is a distance-dependent phenomenon and distances could not be measured for probes more than ~85 Å apart, we did not obtain data for the extended conformation. Likewise, there was no evidence of MR complexes containing dissociated Mre11 dimers (*Käshammer et al., 2019*; *Saathoff et al., 2018*) since the LRET probes on the Rad50 NBDs would also be too far apart in that scenario. Interestingly, this is not the first time that conformations with these names have been suggested for MR. Tainer and coworkers (*Williams et al., 2011*) found that a combination of molecular dynamics (MD)-simulated conformational models best described their ATP-free and ATP-bound Pf MR$^{NBD}$ SAXS curves. Although those MD models do not resemble the models presented here, they nevertheless opened the door for conformational heterogeneity in MR, a possibility also proposed by others (*Deshpande et al., 2014*; *Lafrance-Vanasse et al., 2015*).

Our characterization of Mre11-Rad50 interaction mutants in full-length MR complexes, along with the 41 Å separation between Rad50 monomers, suggests that only the open conformation can accommodate dsDNA in a productive orientation. When the stability of both the open and partially open conformations is compromised (K277E/K279E), MR cannot cleave the Exo2 substrate without ATP, which once hydrolyzed likely results in destabilization of the closed conformation for enough time to allow DNA to bind. Likewise, this mutant had the worst endonuclease activity, suggesting that the partially open conformation also plays a role in endonuclease function. The R177E/E181R mutant only disrupts the open conformation, but unlike K277E/K279E, there is some residual Exo2 activity without ATP. Perhaps this is because the partially open conformation can form and occasionally flexes open enough for dsDNA to bind. This mutant shows reasonable endonuclease activity, though, suggesting that the partially open conformation does allow room for ssDNA. E152K, which also only disrupts the open, has WT-like endonuclease activity providing further evidence for a role of the partially open state in endonuclease activity. Notably, neither MR K227E/K279E nor MR R177E/E181R that should form more of the closed conformation increases ATP hydrolysis; however, this is expected since that reaction is highly regulated by allostery within Rad50 (*Boswell et al., 2018*; *Deshpande et al., 2014*;

*Williams et al., 2011*). On the other hand, D313K, which destabilizes both the closed and partially open conformations to form the open conformation that binds dsDNA, has the highest levels of nuclease activity and cleaves the Exo2 substrate readily without ATP. Additionally, this mutant is able to create the shortest cleavage products from the substrate DNA even though it has significantly decreased ATP hydrolysis, suggesting that although hydrolysis is required to release the closed state and redistribute the population of the conformations, it may not be specifically required for nuclease activity. This hypothesis is further supported when looking at all of the Exo2 reactions without ATP (i.e., $Mg^{2+}$/$Mn^{2+}$): with the exception of the mutant that has a stabilized closed form (K277E/K279E), all of the MR complexes can cleave DNA. Furthermore, the D313K mutant is in a loop that may act as a pivot between the closed and partially open conformations. By hampering the ability of MR to pivot here, there may be more interconversion between partially open and open conformations, which, we hypothesize, drives MR DNA cleavage, be it sequential endo- and exonuclease action or processive exonuclease activity, along the length of the DNA.

In order for the Mre11 dimer to accommodate the Rad50 dimer in the closed conformation, the capping domain has flexed outward ~5 Å (*Figure 2E*, *Figure 2—video 1*) from its position in the partially open and open conformations. Additional capping domain motions are also observed when transitioning between partially open and open conformers. This movement within the capping domain is reminiscent of motions previously observed in structures of the Pf Mre11 dimer bound to DNA substrates that mimicked either a DNA DSB or a stalled replication fork (*Williams et al., 2008*). Thus, our MR models further confirm that motions within the Mre11 capping domain coupled with the H17 wedge residue are important in DNA unwinding and nuclease activity.

In summary, our data suggest a model for MR activity where the ATP-bound closed, partially open, and open conformations exist in equilibrium (*Figure 4C*). Initial recognition of the DNA DSB occurs with the open state, where at least the first two nucleotides can be excised. Resection proceeds as MR cycles between the open and partially open conformations. Since the D313K mutant can perform extensive nuclease activity with very low ATP hydrolysis activity, we suggest that the closed form may not be necessary for exonuclease function per se. Instead, the closed form could serve to reset the equilibrium of the three states once bound ATP is hydrolyzed and products are released. If the free energy for the conformational changes between the partially open and open states is not driven by Rad50 ATP binding and hydrolysis, where else could it come? Richardson and coworkers measured the release of –5.3 kcal/mol (–22.17 kJ/mol) of free energy for the hydrolysis of a DNA phosphodiester bond (*Dickson et al., 2000*). Thus, the free energy released from the Mre11 nuclease activity could be the driving force for the conformational changes proposed here. Importantly, the closed conformation must form in order for MR to be functional as it is required for downstream signaling through ATM. Our model is supported by several recently reported cancer-associated MR DNA DSB separation-of-function mutants that maintain Mre11 nuclease function while losing the ability to signal the presence of the DNA DSB through ATM or vice versa (*Al-Ahmadie et al., 2014*; *Chansel-Da Cruz et al., 2020*; *Hohl et al., 2020*). Thus, the conformational heterogeneity we describe here has important functional consequences for DNA DSB repair.

## Materials and methods
### Plasmids, protein expression, and purification
Plasmids for *P. furiosus* full-length Rad50, Rad50 NBD (Rad50[NBD]: aa1-195; GGAGGAGG linker; aa709-882), and Mre11 protein expression in *E. coli* were previously described (*Boswell et al., 2020*; *Boswell et al., 2018*). Point mutations were introduced using a modified QuikChange (Strategene) approach and were verified by Sanger sequencing. Protein expression and purification were performed as previously described for full-length Rad50 (*Boswell et al., 2020*), Rad50[NBD] (*Boswell et al., 2018*), and Mre11 (*Rahman et al., 2020*).

### ATP hydrolysis assay
Rad50 ATP hydrolysis assays were performed essentially as described by *Boswell et al., 2018*. 0–300 µM ATP was titrated into microfuge tubes containing either 2.5 µM MR[NBD] complex or 2 µM full-length MR complex and 50 mM Tris, 80 mM NaCl, 1% glycerol, 5 mM $MgCl_2$, pH 7. Reactions without protein were included for each ATP concentration to control for ATP degradation and $PO_4$

contamination. 60 µL reactions were incubated at 65°C for 60 min after which the tubes were placed on ice. 50 µL of each reaction was then transferred to the wells of clear, flat-bottom 96-well plates, and 100 µL of cold BIOMOL Green (Enzo Lifesciences) colorimetric reagent was added. After a 30 min incubation at room temperature to allow the color to develop, the amount of inorganic phosphate released by hydrolysis was quantified using the absorbance mode on a Synergy Neo2 multi-mode plate reader. BIOMOL Green signal ($A_{640}$) was corrected by subtracting the $A_{640}$ values of the ATP-only reactions at each ATP concentration and then transformed into pmols of $PO_4$ released/min based on a $PO_4$ standard curve incubated in BIOMOL Green reagent for 30 min at room temperature. Plots of $PO_4$ released/min ($v_0$) versus ATP concentration were fit to the Michaelis–Menten equation including a Hill coefficient (n).

$$v_0 = \frac{V_{max}\left[ATP\right]^n}{K_M^n + \left[ATP\right]^n} \tag{1}$$

## 384-Well plate nuclease assays

Nuclease activity was assayed by monitoring the fluorescence of a 2-aminopurine (2-AP) nucleotide incorporated into a dsDNA substrate. 2-AP fluorescence is quenched within the base-pair, and release of the 2-AP from the duplex by Mre11 nuclease activity results in an increase in the 2-AP fluorescence signal. The Exo2 substrate was formed by annealing equimolar amounts of 5′-GGCGTGCCTTGGGCGCGCTGCGGGCGG[2-AP]G-3′ and 5′-CTCCGCCCGCAGCGCGCCCAAGGCACGCC-3′ DNA oligos (IDT). For LRET Rad50$^{NBD}$ cysteine mutants, 30 µL exonuclease reactions were mixed in microfuge tubes and contained 0.5 µM MR$^{NBD}$ complex and 1 µM Exo2 dsDNA substrate in 50 mM HEPES, 100 mM NaCl, 5 mM $MgCl_2$, 0.1 mM EDTA, 1% glycerol, 1 mM TCEP, pH 7. 1 mM $MnCl_2$ was added to the reactions as indicated. After a 45 min incubation at 60°C, the tubes were removed from the heat block and spun. 25 µL of each reaction was transferred to black, flat-bottom 384-well plates and 2-AP fluorescence was quantified by the Synergy Neo2 plate reader (ex310/em375). For the Mre11-Rad50 interaction mutants, exonuclease activity was assessed for MR complexes made with full-length Rad50. Reactions with 0.5 µM full-length MR complex and 1 µM Exo2 were assembled as above, except 1 mM $MnCl_2$, 1 mM $MnCl_2$/1 mM ATP, or 1 mM $MnCl_2$/1 mM AMPPNP were added as indicated and reactions were incubated at 60°C for 15 min. In addition to the Exo2 substrate, a second substrate, Exo11, was also employed for full-length MR: 5′-GGCGTGCCTTGGGCGCGC[2-AP]GCGG-GCGGAG-3′ annealed to 5′-CTCCGCCCGCTGCGCGCCCAAGGCACGCC-3′ (IDT). Exo11 reactions were incubated at 60°C for 30 min. Endonuclease activity assays of the various Mre11-Rad50 interaction mutants used a 17-nucleotide ssDNA substrate containing a fluorophore and quencher pair (5′-Cy3-TCTCTAGCAGTGGCGCC-BHQ2-3′; IDT), based on a previously reported assay (*Yuan et al., 2016*). Once the ssDNA is cleaved, the fluorophore and quencher separate, allowing Cy3 fluorescence to be detected. Reactions with full-length MR complexes were performed just as the exonuclease assays described above, except 0.2 µM Cy3/BHQ2 substrate was used. Endonuclease reactions were incubated for 30 min at 60°C. Cy3 fluorescence was then quantified by the Synergy Neo2 plate reader (ex535/em570). Data for each of the described nuclease assays were collected in triplicate.

## Gel-based nuclease assays

Nuclease activity of full-length MR complexes on four separate substrates was analyzed on denaturing polyacrylamide gels: a 40-nucleotide ssDNA substrate (5′-Cy3-GTGTTCGGACTCTGCCTCAAGACGGTAGTCAACGTGCTTG-3′; IDT), a 40-nucleotide dsDNA substrate (the ssDNA substrate annealed to an unlabeled complementary strand), a 36-nucleotide DNA hairpin (5′-FAM-CACGCACGTAGAAGCTTTTGCTTCTACGTGCGTGAC-3; IDT, containing a 15-base-pair helix and a 2-nucleotide 3′-overhang), and a 50-nucleotide dsDNA where the labeled strand has phosphorothioate bonds between the six nucleotides at the 3′-end (5′-Cy5-CTGCAGGGTTTTTGTTCCAGTCTGTAGCACTGTGTAAGACAGGCCsAsGsAsTsG-3′, annealed to 5′- CATCTGGCCTGTCTTACACAGTGCTACAGACTGGAACAAAAACCCTGCAG-3′; IDT). Nuclease reactions in microfuge tubes contained 0.5 µM full-length MR complex and 1 µM DNA substrate in 50 mM HEPES, 100 mM NaCl, 5 mM $MgCl_2$, 1 mM $MnCl_2$, 0.1 mM EDTA, 1% glycerol, 1 mM TCEP, pH 7. 1 mM ATP was added to the reactions as indicated. After a 45 min incubation at 60°C, 1.5 µL of each reaction was transferred to 28.5 µL of the above buffer with 20 mM EDTA. These samples were then mixed with 30 µL of loading buffer (8 M urea, 20 mM EDTA, 6% Ficoll 400) before 20 µL was loaded onto 15% denaturing polyacrylamide gels in 1×

TBE buffer. Gels were run at 3 hr at a constant power of 20 W and scanned by an Amersham Typhoon 5 fluorescence imager. FAM-labeled substrates were imaged with a 488 nm laser and 525BP20 filter, Cy3-labeled substrates with a 532 nm laser and 570BP20 filter, and Cy5-labeled substrates with a 635 nm laser and 670BP30 filter. The images were analyzed using ImageQuant software.

## Labeling Rad50 cysteine mutants with LRET probes

Thiol-reactive $Tb^{3+}$ chelate DTPA-cs124-EMPH (LanthaScreen, Life Technologies, Inc) was used as the LRET donor for all samples and Bodipy FL maleimide (Invitrogen) or Cyanine3 (Cy3) maleimide (GE Healthcare) were used as acceptor fluorophores. 50 µM purified $Rad50^{NBD}$ containing a single cysteine was combined with a twofold excess of one of the labels in degassed Labeling Buffer (25 mM Tris, pH 8, 100 mM NaCl, 10% glycerol, 1 mM TCEP) in 100 µL reactions. The reaction was incubated in the dark at room temperature for 1.5–2 hr. Labeling of full-length Rad50 cysteine mutants followed the same protocol, except the donor and one acceptor label were added simultaneously to each reaction (twofold molar ratio of each). Unreacted label was removed by running the labeling reaction over a Superdex 200 Increase 10/300 GL column (GE Healthcare) equilibrated with 25 mM HEPES, 200 mM NaCl, 10% glycerol, 1 mM TCEP, pH 7. Eluted protein was concentrated using centrifugal concentrators (VivaSpin, Sartorius) and then assessed for successful labeling by measuring fluorescence using the following settings and the monochromator in the BioTek Synergy Neo2 plate reader: ex337/em490 for $DPTA-Tb^{3+}$, ex485/em515 for Bodipy FL, and ex535/em570 for Cy3. Concentrated labeled proteins were aliquoted, frozen in liquid nitrogen, and stored at –80°C.

## LRET data collection and analysis

For LRET experiments, 40 µL mixtures of 1 µM Mre11 + 0.5 µM $Tb^{3+}$-labeled $Rad50^{NBD}$ + 0.5 µM Bodipy- or Cy3-labeled $Rad50^{NBD}$ were heated at 60°C for 15 min to form the $MR^{NBD}$ complex. Full-length MR LRET mixtures were 1 µM Mre11 + 1 µM dual-labeled (donor + acceptor) Rad50. Each LRET sample was subsequently diluted to 160 µL (0.25 µM complex) with LRET buffer (50 mM HEPES, 100 mM NaCl, 5 mM $MgCl_2$, 0.1 mM EDTA, 1% glycerol, 1 mM TCEP, pH 7). Next, for samples containing nucleotide, 2 mM of either ATP or ATPγS was added and the samples were heated again for 20 min at 50°C. Finally, for samples containing DNA, either 34 µM hairpin DNA (5'-CACGCACGTAGAAGCT TTTGCTTCTACGTGCGTGAC-3') or 16.6 µM ssDNA (5'-TGTAGTGCATTGCGTTTTTGCTTCTACG TGCGTGAC-3') was added to ATP-bound LRET samples and heated once more for 15 min at 50°C. These concentrations of DNA should give >95% bound complex in LRET conditions at 50°C. 150 µL were transferred into a 3 mm pathlength cuvette, and LRET data were collected after a final 5 min incubation in the fluorimeter at 50°C. Donor and acceptor lifetimes were calculated from intensity decays measured with an Optical Building Blocks phosphorescence lifetime photometer (EasyLife L). LRET samples were excited through a narrow band 335 nm filter (Semrock FF01-335/7) and after a 200 µs delay (to allow the decay of sample autofluorescence, emission due to direct excitation of the acceptor, and scattering of the excitation pulse), donor emission intensity was collected for $Tb^{3+}$ at 50 Hz through a 490/10 nm band-pass filter (Omega Optical), and donor-sensitized acceptor emission intensities were collected at 100 Hz through 520/10 nm (Bodipy FL) or 570/10 nm (Cy3) band-pass filters. PTI FeliX32 software was used to fit the Bodipy FL and Cy3 emission decay curves to either a two- or three-exponential function depending on the identity of the LRET pair. For samples where the distance between the LRET pair in the nucleotide-bound crystal structure was ≤42 Å, the data fit well to three exponentials for both Bodipy- and Cy3-labeled samples. For samples where the LRET pair was separated by more than 45 Å, the Bodipy emission decays fit to only two exponentials while the Cy3 decays fit to three. Lifetime distributions shorter than ~100 µs were discarded as these are largely a function of the instrument response time (*Cooper and Altenberg, 2013*; *Zoghbi et al., 2017*; *Zoghbi et al., 2012*; *Zoghbi and Altenberg, 2018*). Donor $Tb^{3+}$-chelate fluorescence decays were recorded at each probe position in donor-only-labeled $MR^{NBD}$ complexes. Each $Tb^{3+}$-chelate fluorescence decay fit well to two exponentials, with the longer lifetime comprising >85% of the signal. This longer lifetime was used in the distance analysis. From the $Tb^{3+}$-chelate, Bodipy FL, and Cy3 lifetimes, the distances between donor and acceptor molecules (R) were calculated with

$$E = 1 - \frac{\tau_{DA}}{\tau_D} \qquad (2)$$

and

$$R = R_0 \left( E^{-1} - 1 \right)^{1/6} \tag{3}$$

where E is the efficiency of energy transfer, $\tau_{DA}$ is the donor-sensitized lifetime of the acceptor (Bodipy FL or Cy3), $\tau_D$ is the lifetime of the donor ($Tb^{3+}$), and $R_0$ is the Förster distance between $Tb^{3+}$ and Bodipy FL (44.9 Å) or $Tb^{3+}$ and Cy3 (61.2 Å). Errors are the standard deviations from the mean of at least three measurements.

## HADDOCK

Molecular docking of the MR^NBD complex was done using the GURU interface on the HADDOCK 2.4 webserver (*Dominguez et al., 2003*; *van Zundert et al., 2016*). For the three-body docking protocol, the PDB inputs were 3DSC (Pf Mre11 dimer lacking the helix-loop-helix [HLH] motifs and C-termini, DNAs deleted) (*Williams et al., 2008*) and two monomers of 3QKU (AMPPNP-bound Pf Rad50^NBD in complex with the Mre11 HLH motif) (*Williams et al., 2011*). Except for increasing the number of structures for rigid body docking (it0) to 3000, all settings used were default. The three linkers attaching the Pf Mre11 capping domain to the nuclease domain were allowed to be fully flexible (Y222-V236, Y249-G254, and V266-F273). To allow HADDOCK to move the two Rad50 monomers relative to each other, we input Rad50 as two identical monomers. C2 symmetry was enforced between the two Rad50^NBD monomers and between the two monomers of the Mre11 dimer.

HADDOCK Mre11 to Rad50 active ambiguous interaction restraints (AIRs) were based on *M. jannaschii* (3AV0) (*Lim et al., 2011*) and *E. coli* (6S6V) (*Käshammer et al., 2019*) ATP-γ-S-bound MR structures. Passive AIRs (solvent accessible surface neighbors of active residues) were automatically defined by HADDOCK based on these active AIRs. Two extra passive AIRs were included on the 'back side' of the Mre11 capping domain to allow for the extended structure seen in ATP-free *T. maritima* MR (PDB: 3QG5; *Lammens et al., 2011*). *Figure 1—figure supplement 2* shows the location of these AIRs on the structures of Mre11 and Rad50. The same set of active and passive AIRs were used in all of the HADDOCK runs with 50% random exclusion of AIRs in each structure calculation. The measured LRET distances were input as unambiguous restraints and defined as the Cβ-Cβ distance between the LRET-labeled residues, ± 5 Å. For distance restraints greater than 75 Å, the bounds were increased to ±7 Å as the lifetime fits giving these distances were in a more error-prone region (i.e., the flat section) of the $Tb^{3+}$-Cy3 LRET efficiency curve. The model with the lowest HADDOCK score in each run was considered as the best structure. To ensure that the unambiguous distance restraints obtained from one probe position were not dominating the structure calculations, HADDOCK runs were performed with systematic dropouts of all restraints calculated from a specific cysteine position (*Figure 2—figure supplement 2*). For the closed model, all of the dropout structures maintained the proper D-loop/Walker A juxtaposition. PyMOL version 2.4 was used to calculate RMSD values for all-atom alignments of HADDOCK models and to make figures of structures.

## SAXS and FoXS analysis

MR^NBD samples for SAXS were made by mixing Mre11 and Rad50^NBD in a 1:1.1 molar ratio. Samples were heated at 60°C for 15 min to form complex and then cooled on the benchtop. The complex was then loaded onto a HiLoad 16/60 Superdex 200 column (GE Healthcare) equilibrated in 25 mM HEPES, 0.2 M NaCl, 0.1 mM EDTA, 1 mM TCEP, pH 7. The MR^NBD peak was collected and concentrated (Vivaspin, Sartorius) before dialysis (Slide-A-Lyzer MINI, Thermo Scientific) into LRET buffer (50 mM HEPES, 100 mM NaCl, 5 mM $MgCl_2$, 0.1 mM EDTA, 1% glycerol, 1 mM TCEP, pH 7) ± 2 mM ATP overnight at room temperature. Samples were diluted to 4 mg/mL (~22 μM MR^NBD complex) with the equilibrated dialysis buffer and sent to the SIBYLS beamline 12.3.1 at the Advanced Light Source in Berkeley for high-throughput SAXS analysis (*Classen et al., 2013*; *Dyer et al., 2014*; *Hura et al., 2009*). For each sample, a total of 33 frames were collected at 0.3 s intervals at 10°C with a cell thickness of 1.5 mm. The sample to detector distance was 2 m, and the beam wavelength was 11 keV/1.27 Å. Buffer profiles were subtracted from sample profiles, and the buffer-subtracted frames for each sample were averaged using SAXS FrameSlice (https://sibyls.als.lbl.gov/ran).

Compared to the 3QKU PDB used as the input for HADDOCK runs, the Rad50^NBD protein construct used in the LRET and SAXS samples has an additional seven amino acids on one side of the truncated

coiled-coils and eight amino acids on the other and the linker (GGAGGAGG) connecting them is slightly longer. To ensure that the structural models were as close to the protein construct used for the SAXS experiments as possible, the Rad50 coiled-coil and linker along with the 14 amino acid loop connecting the Mre11 helix-loop-helix and nuclease domain were modeled with loop modeling via Rosetta (version 3.11) (*Mandell et al., 2009*; *Wang et al., 2007*). This resulted in models only missing the 47 C-terminal amino acids of Mre11 as compared to the experimental protein (*Figure 4—figure supplement 1*). The FoXS server was used to back-calculate SAXS profiles of these modified closed, partially open, and open HADDOCK-Rosetta models and to fit these to the experimental $MR^{NBD}$ scattering curves. The MultiFoXS server was used to calculate population-weighted ensemble fits (*Schneidman-Duhovny et al., 2016*; *Schneidman-Duhovny et al., 2013*).

## Acknowledgements

We thank Dr. Mariana Fiori and Prof. Guillermo Altenberg (TTUHSC, Lubbock, TX) for use of the fluorimeter for LRET data collection, technical help, and suggestions. We also thank Dr. Greg Hura (SIBYLS) for helpful suggestions. SAXS data was collected at SIBYLS, which is supported by the DOE-BER IDAT DE-AC02-05CH11231 and NIGMS ALS-ENABLE (P30 GM124169 and S10OD018483). We also thank the National Institutes of Health (grant number: 1R35GM128906), CPRIT (grant number: RP180553), and The Welch Foundation (grant number: D-1876) for financial support to MPL.

## Additional information

### Funding

| Funder | Grant reference number | Author |
| --- | --- | --- |
| National Institute of General Medical Sciences | 1R35GM128906 | Michael Latham |
| Cancer Prevention and Research Institute of Texas | RP180553 | Michael Latham |
| Welch Foundation | D-1876 | Michael Latham |

The funders had no role in study design, data collection and interpretation, or the decision to submit the work for publication.

### Author contributions

Marella D Canny, Conceptualization, Formal analysis, Investigation, Methodology, Visualization, Writing - original draft, Writing – review and editing; Michael P Latham, Conceptualization, Funding acquisition, Project administration, Supervision, Visualization, Writing – review and editing

### Author ORCIDs

Marella D Canny (iD) http://orcid.org/0000-0002-9884-5575
Michael P Latham (iD) http://orcid.org/0000-0002-2209-5798

### Decision letter and Author response

Decision letter https://doi.org/10.7554/eLife.69579.sa1
Author response https://doi.org/10.7554/eLife.69579.sa2

## Additional files

### Supplementary files

• Supplementary file 1. $MR^{NBD}$ luminescence resonance energy transfer (LRET) probe pair distances. Each row represents data from a unique LRET pair. Distances (in Å) were calculated from the decay of donor-sensitized Bodipy or Cy3 fluorescence emission as described in Materials and methods. Errors are the standard deviation of n ≥ 3 LRET measurements.

• Supplementary file 2. $MR^{NBD}$ luminescence resonance energy transfer (LRET) experimental probe distances and HADDOCK model distances. The average experimental LRET distance measured

for each pair of Rad50 LRET probes was used as unambiguous restraints in closed, partially open, and open HADDOCK simulations. The three resulting HADDOCK models had the reported Cβ-Cβ distance between indicated probe positions.

• Transparent reporting form

### Data availability

All data generated or analysed during this study are available from the DRYAD database under the doi: https://doi.org/10.5061/dryad.qfttdz0h6.

The following dataset was generated:

| Author(s) | Year | Dataset title | Dataset URL | Database and Identifier |
|---|---|---|---|---|
| Canny MD, Latham MP | 2021 | Data from: LRET-derived HADDOCK structural models describe the conformational heterogeneity required for processivity of the Mre11-Rad50 DNA damage repair complex | http://dx.doi.org/10.5061/dryad.qfttdz0h6 | Dryad Digital Repository, 10.5061/dryad.qfttdz0h6 |

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
