## [Editor Report]

This study on the Mre11 and Rad50 proteins is of interest to biologists studying DNA repair. Advances in the understanding of how structural states of Mre11-Rad50 complex are linked to DNA end detection and DNA processing, as addressed in this study, are of central importance to research on genome stability and DNA repair, with implications in human disease such as cancer and immune disorders. Enzymatically, RAD50 is an ATPase and MRE11 is a nuclease with both exo- and endonuclease activities. How all these functions are catalyzed by the complex remains unresolved. Through the combination of biophysical analyses and biochemical activity assays, this study identifies three conformations of ATP-bound *P. furiosus* Mre11-Rad50 complex – open, partially open, and closed – and links these activities to the Mre11-Rad50 function.

---

## [Decision Letter]

**Decision letter after peer review:**

Thank you for submitting your article "LRET-derived HADDOCK structural models describe the conformational heterogeneity required for processivity of the Mre11-Rad50 DNA damage repair complex" for consideration by *eLife*. Your article has been reviewed by 3 peer reviewers, and the evaluation has been overseen by Maria Spies as the Reviewing Editor and Cynthia Wolberger as the Senior Editor. The reviewers have opted to remain anonymous.

Essential revisions:

1. Include DNA in the structural studies to determine whether the observed states, and in particular the new state, is relevant to the complex "in action" bound to DNA where quite different conformations might occur.

2. Use the more relevant endonuclease activity to validate the mutants.

3. Repeat the experiments with the non-hydrolysable ATP analogue AMP-PNP, and in the absence of nucleotide

*Reviewer #2 (Recommendations for the authors):*

1) LRET measurements were performed under ATP-saturating conditions, yet it provided conformations compatible with Apo-MR(NBD) (see below). It would be good to repeat these experiments with the non-hydrolysable ATP analogue AMP-PNP, and in the absence of nucleotide. These experiments and the subsequent analysis using HADDOCK and SAXS shall provide insights into the coupling of ATP binding and hydrolysis with conformational changes of the MR complex.

2) It is unclear how the open or close conformation SAXS profiles modelled with FOXS software and their fitting to the SAXS data of MR(NBD) samples with a without ATP confirm the three conformations of ATP-bound MR(NBD). The quality of the fittings was almost identical considering either two or three populations of conformational states. Thus, the statement "the SAXS data supports the three states observed in LRET data" is not accurate. These experiments do not exclude the possibility of a third state, but does not support it either. This should be changed.

3) The region with largest discrepancy between SAXS (WT apo) and model (Open) data are in the range of 0.1-0.2 Angstroms. Could the authors elaborate on the reasons why is this?

4) It might be instructive to include the modelled data taking into account the percentages of different populations in the two-state model and the three-state model.

5) Another point relates to the stability of the "partially-open" state. Could it just be reflecting the dynamics of the transition between open and close conformations? Authors should discuss on this.

6) Along this line, how is it possible that the open state FoXS-calculated theoretical SAXS curve derived from the HADDOCK model obtained under saturating ATP-binding conditions mostly matches with the apo SAXS data (Figure 4A)?

Other concerns:

1) Authors should specify the reason for choosing experimental conditions at 50°C, which might not be obvious for the non-specialized reader.

2) Along this line, it would be good to make the distinction between LRET and FRET, and the reasons to choose the former to study protein-protein interactions.

3) Table S1. There are some data missing, such as S13C (Bodipy position) and S13C (TB3+ position). Also some values in the table do not match with Figure 1D; for instance, S93C (Cy3) and S93C (TB3+). Please revise the entire table for consistency.

4) Since the authors evaluate the possibility for DNA to interact with the nuclease active site of Mre11 and how the different conformations of the MR complex might occlude the access to the site, I suggest to include the nuclease active site of MRE11 in their models of Figure 2.

5) Figure S1B. I guess ATP is included in the reaction. This should be indicated in the text or caption.

*Reviewer #3 (Recommendations for the authors):*

My main suggestion would be to focus also on the endonuclease activity. The authors only analyze the exonuclease, which is, at least in part, dependent on RAD50 in P. furiosus. However, in eukaryotes MRE11 is fully active as an exonuclease without RAD50, raising questions to which extent the P. furiosus is a useful model in this regard. However, as shown previously (Hopkins and Paull, Cell), also MR from P. furiosus exhibits endonuclease activity. The endonuclease is important for recombination and is in particular dependent on RAD50 in all organisms including eukaryotes. Therefore, I suggest to investigate the constructed mutants also in terms of their endonuclease activity, using assays such as established in Hopkins and Paull. In my opinion, it would be a breakthrough to observe whether the different conformations affect the exo- versus endonuclease activities of MR.

Other comments:

Figure S1: It would be helpful to include negative controls (ATPase dead, or nuclease-dead) MR variants to unambiguously link the observed activity with the recombinant construct.

My understanding is that the coiled-coil of Rad50 in eukaryotes is essential for all its biological functions (Petrini laboratory). The authors used the truncation variant and observed ATPase and exonuclease activities. Is there is a difference between the full-length and truncated MR variants in these assays?

How do the authors know that all conformations observed are ATP-bound?

---

## [Author Response]

Essential revisions:1. Include DNA in the structural studies to determine whether the observed states, and in particular the new state, is relevant to the complex "in action" bound to DNA where quite different conformations might occur.

We thank the Reviewers for this suggestion. We have now collected LRET data on ATP-bound MR^NBD^ in the presence of a hairpin dsDNA or a ssDNA as substrates. Overall, the same three conformations were observed with both substrates. No significant differences in distances between the Rad50 protomers were observed in the presence of the hairpin dsDNA. However, the ssDNA appeared to have a slightly narrower open conformation (i.e., shorter distances between Rad50 protomers) in each of the LRET pairs tested. This observation was true for MR^NBD^ and full-length MR complexes. The differences observed, however, were too small to necessitate re-running the HADDOCK simulations as nearly all of them were within the ±7 Å bounds set on the ‘open’ restraints. This new data is presented in a new figure (Figure 4) and in a new sub-section within the Results section (Multiple conformations of MR persist for various substrate-bound states; pgs. 14-16) of the modified manuscript. We also added a linearized plasmid dsDNA and did not observe any effects in either MR^NBD^ or full-length MR constructs (data not shown).

As we note on pg. 15 of the revised manuscript, the result that DNA substrates do not change the observed conformations of Rad50 protomers within the MR complex is not surprising and does not preclude the existence of the recently proposed “cutting” state observed by cryo-EM on *E. coli* SbcCD. We cannot say anything about the relative orientations of Rad50 protomers to Mre11 from the LRET data presented here because our LRET probes only report on the distances between Rad50s. Indeed, from the perspective of the Rad50 protomers, the “cutting” state is “closed” and would therefore be consistent with our data.

2. Use the more relevant endonuclease activity to validate the mutants.

We thank the Reviewers for this suggestion; this gave us the push to get the gel-based assays running in the lab. As can be seen in the modified manuscript, we dramatically expanded the nuclease data presented in the validation mutants section. We now include a fluorescence 384-well plate-based ssDNA endonuclease assay (Figure 3D) along with the existing Exo2 (Figure 3B) and Exo11 (Figure 3C) data. These assays were complemented with nuclease assays resolved by denaturing PAGE using a 40-mer dsDNA, a hairpin DNA containing a two nucleotide 3’-end overhang, a 40-mer ssDNA, and a 50-mer dsDNA where the terminal five phosphodiester bonds on the labeled strand are replaced by phosphorothioate bonds (making this substrate incompatible for 3’-to-5’ exonuclease activity on that strand). The gel-based data is presented in new Supplemental Figure S4. Most of these gel-based nuclease activity assays were performed both in the presence and absence of saturating ATP.

As described on pgs. 10-12 of the revised manuscript, we found that the Exo2 data compared favorably with the 40-mer dsDNA cleavage in the absence of ATP resolved by the PAGE – the 3’-end nucleotide was removed without further cleavage. The Exo11 results of each mutant MR complex resembled those of the gel-based 40-mer dsDNA and hairpin DNA cleavage in the presence of ATP – combined endo- and exonuclease activities by Mre11 resulted in a ladder of DNA products. Lastly, the ssDNA endonuclease assay results were similar to the data from both the 40-mer ssDNA and the 3’-end phosphorothioate-blocked dsDNA substrates – here, endonuclease activity also resulted in products of different sizes.

Several interesting observations arose from these data. First, the Exo11 assay is actually an endo/exo assay. Second, we noted that the previously described Pf Mre11 exonuclease-deficient mutant H52S does indeed have exonuclease activity. When assayed with the Exo2 substrate in the absence of ATP, as previously described (Williams, R.S., et al. 2008 Cell 135:97–109), we observed no cleavage; however, the gel-based assays clearly show nuclease activity against all the substrates tested. Second, H17E did not cleave either the dsDNA 40-mer or hairpin DNA, as it is unable to unwind the double helix, but showed robust activity on the phosphorothioate-blocked dsDNA and on the ssDNA. This suggests the primary cleavage of the phosphorothioate-blocked substrate is via endonuclease activity. Finally, the new nuclease data seem to solidify that ATP binding, but not hydrolysis, is required for sequential or processive nuclease activity. On dsDNA, leaving ATP out of the reaction results in cleavage that does not proceed past the 3’ base pair. We hypothesize that ATP binding alters the structure of Rad50 which may affect its ability to bind DNA in the partially open and open forms. This is an idea that we are eager to address in follow-up studies.

3. Repeat the experiments with the non-hydrolysable ATP analogue AMP-PNP, and in the absence of nucleotide

We also collected LRET data in the absence of nucleotide or with the non-hydrolysable analog ATPγS, which in our NMR experience is more stable at 50 °C than AMPPNP. Overall, we observed the same three conformations of MR^NBD^ and full-length MR as for the ATP-bound form. This data is presented in the new Figure 4 and in the new Results sub-section (pgs. 14-16). Although for several of the pairs, the apo open distance was 2-4 Å larger (wider) than nucleotide-bound, the distances did not change significantly, and we did not re-run the HADDOCK calculations and SAXS experiments. These results are not surprising – given the size of the fluorophores and their associated linkers, LRET-based distances do not provide high enough resolution to report on the small, local conformational changes, such as those reported by Tainer and co-workers (Williams, G.J., et al. 2011 Nat. Struct. Mol. Biol. 18:423–431), within Rad50^NBD^ that occur upon ATP binding and subsequent hydrolysis.

What is likely changing within the nucleotide-bound forms is the relative populations of the three observed conformations. Unfortunately, the calculation of populations is very difficult from our LRET measurements. In theory, population information should be contained in the pre-factors of the multi-exponential fit. However, because neither LRET pair is sensitive to all three conformations (i.e., the protomers in open are too far apart for energy transfer with Tb^3+^-BODIPY and they are too close for Tb^3+^-Cy3), we cannot obtain accurate populations from these data.

Reviewer #2 (Recommendations for the authors):1) LRET measurements were performed under ATP-saturating conditions, yet it provided conformations compatible with Apo-MR(NBD) (see below). It would be good to repeat these experiments with the non-hydrolysable ATP analogue AMP-PNP, and in the absence of nucleotide. These experiments and the subsequent analysis using HADDOCK and SAXS shall provide insights into the coupling of ATP binding and hydrolysis with conformational changes of the MR complex.

As described in more detail above (Essential Revision #3), we have collected LRET data in the absence of nucleotide or with the non-hydrolysable analog ATPγS (more stable at 50 °C than AMP-PNP), and we observe the same three conformations. As described above, we have added to the Results section of the modified manuscript (pgs. 14-16) and included a new figure (Figure 4) describing these data.

2) It is unclear how the open or close conformation SAXS profiles modelled with FOXS software and their fitting to the SAXS data of MR(NBD) samples with a without ATP confirm the three conformations of ATP-bound MR(NBD). The quality of the fittings was almost identical considering either two or three populations of conformational states. Thus, the statement "the SAXS data supports the three states observed in LRET data" is not accurate. These experiments do not exclude the possibility of a third state, but does not support it either. This should be changed.

We appreciate this comment. We have modified the indicated sentence to read that “the SAXS data supports the presence of multiple states as observed in LRET data” (pg. 16).

3) The region with largest discrepancy between SAXS (WT apo) and model (Open) data are in the range of 0.1-0.2 Angstroms. Could the authors elaborate on the reasons why is this?

As seen in the update to the figure made from the suggestion in the point below (now Figure 5A, right), visibly better fits are obtained in the 0.1 – 0.2 Å^-1^ range when multiple models are used to back-calculate the scattering data. Thus, we would interpret the noted deviation as being indicative of the presence of multiple conformations in solution. We have made note of the deviation and explained it (pg. 16) in the modified text.

4) It might be instructive to include the modelled data taking into account the percentages of different populations in the two-state model and the three-state model.

We thank the Reviewer for this suggestion. Figure 5 (original Figure 4) has been updated with an additional panel showing the two-state fits to the apo and ATP SAXS data.

5) Another point relates to the stability of the "partially-open" state. Could it just be reflecting the dynamics of the transition between open and close conformations? Authors should discuss on this.

Donor-sensitized acceptor lifetimes used to calculate the distances of the partially open state are ~0.5 ms. For this state to be observable by LRET, it must have a lifetime longer than the timescale of the measurement, which puts a lower limit on the lifetime of the partially open state at ~5 ms. We have added a statement regarding this to the modified Discussion (pg. 16).

6) Along this line, how is it possible that the open state FoXS-calculated theoretical SAXS curve derived from the HADDOCK model obtained under saturating ATP-binding conditions mostly matches with the apo SAXS data (Figure 4A)?

We currently think that the SAXS data, like the LRET data, does not have the resolution to discern the local changes that occur within the proteins of the MR^NBD^ complex upon ATP-binding, but both methods are quite sensitive to global conformations of the complex, which is why in part we choose to use this technique for validation. This being said, it is possible that an open state HADDOCK model made using an apo Rad50^NBD^ structure (as opposed to the AMPPNP-bound crystal structure used here) may fit the apo SAXS data even better. Nonetheless, we are more concerned here with the global conformations of MR and their role in MR function.

We would also like to note that it is clear from the new panel in Figure 5 (original Figure 4) that the apo SAXS curve contains more than one conformation. In the newly collected LRET data (Essential Revision #3), we show that the three conformations exist in the apo (and DNA-bound) conditions as well.

Other concerns:1) Authors should specify the reason for choosing experimental conditions at 50°C, which might not be obvious for the non-specialized reader.

We have added to the Results (pg. 6) to explain the rationale behind the experimental conditions.

2) Along this line, it would be good to make the distinction between LRET and FRET, and the reasons to choose the former to study protein-protein interactions.

We have also added to the Results (pg. 5) to provide the distinctions between LRET and FRET.

3) Table S1. There are some data missing, such as S13C (Bodipy position) and S13C (TB3+ position). Also some values in the table do not match with Figure 1D; for instance, S93C (Cy3) and S93C (TB3+). Please revise the entire table for consistency.

Thank you. We have made these corrections.

4) Since the authors evaluate the possibility for DNA to interact with the nuclease active site of Mre11 and how the different conformations of the MR complex might occlude the access to the site, I suggest to include the nuclease active site of MRE11 in their models of Figure 2.

We thank the Reviewer for this suggestion. We now show Mre11 H85, which helps to coordinate the catalytic Mn^2+^ ions, as violet spheres and Mre11 H17, which acts as “wedge” to unwind dsDNA, as green spheres in each of the structures in Figure 2.

5) Figure S1B. I guess ATP is included in the reaction. This should be indicated in the text or caption.

We are sorry for the confusion. ATP is not included in this Exo2 assay, and this information has been added to the Supplemental Figure S1B figure legend.

Reviewer #3 (Recommendations for the authors):My main suggestion would be to focus also on the endonuclease activity. The authors only analyze the exonuclease, which is, at least in part, dependent on RAD50 in P. furiosus. However, in eukaryotes MRE11 is fully active as an exonuclease without RAD50, raising questions to which extent the P. furiosus is a useful model in this regard. However, as shown previously (Hopkins and Paull, Cell), also MR from P. furiosus exhibits endonuclease activity. The endonuclease is important for recombination and is in particular dependent on RAD50 in all organisms including eukaryotes. Therefore, I suggest to investigate the constructed mutants also in terms of their endonuclease activity, using assays such as established in Hopkins and Paull. In my opinion, it would be a breakthrough to observe whether the different conformations affect the exo- versus endonuclease activities of MR.

As described in detail above (Essential Revisions #2), we now include plate- and gel-based endonuclease activity assays, using a variety of DNA substrates, for all of the validation mutants. We have expanded Figure 3 and included a new Supplemental Figure S4 to show this data. As described above, we have also expanded the Results section of the modified manuscript (pgs. 10-12) to reflect these findings.

Other comments:Figure S1: It would be helpful to include negative controls (ATPase dead, or nuclease-dead) MR variants to unambiguously link the observed activity with the recombinant construct.

We thank the Reviewer for this nice suggestion. We added Rad50 K36R (Walker A mutant) and Mre11 H52S (inactive mutant) data to Supplemental Figure S1A (Rad50 ATP hydrolysis) and S1B (Mre11 Exo2), respectively.

My understanding is that the coiled-coil of Rad50 in eukaryotes is essential for all its biological functions (Petrini laboratory). The authors used the truncation variant and observed ATPase and exonuclease activities. Is there is a difference between the full-length and truncated MR variants in these assays?

We are sorry for this confusion. All of the nuclease assays (except for the Supplemental Figure S1 which is just for the activity of the LRET cysteine mutants) are done in full-length MR complex. In fact, the Exo11 assay requires full-length MR complex. We have tried to make the figure legends and Methods more clear as to which constructs were used in which assays.

How do the authors know that all conformations observed are ATP-bound?

This is an interesting point. We cannot of course say for absolute certainty that all the observed conformations are ATP-bound, but we are at [ATP] where the Rad50s should be >95% bound according to the observed K_D_ for ATP binding under the assay conditions, as stated on pg. 6 of the manuscript.